# Extreme dust storm over the eastern Mediterranean in September 2015: Satellite, lidar, and surface observations in the Cyprus region

Rodanthi-Elisavet Mamouri[1,2], Albert Ansmann[3], Argyro Nisantzi[1], Stavros Solomos[2], George Kallos[4], and Diofantos G. Hadjimitsis[1]

[1]Cyprus University of Technology, Department of Civil Engineering and Geomatics, Limassol, Cyprus
[2]National Observatory of Athens, Athens, Greece
[3]Leibniz Institute for Tropospheric Research, Leipzig, Germany
[4]University of Athens, School of Physics, Division of Environment and Meteorology, Athens, Greece

*Correspondence to:* R.-E. Mamouri (rodanthi.mamouri@cut.ac.cy)

**Abstract.** A record-breaking dust storm originating from desert regions in northern Syria and Iraq occurred over the Eastern Mediterranean in September 2015. In this contribution of a series of two articles (part 1, observations, part 2, atmospheric modeling), we provide a comprehensive overview about the aerosol conditions during this extreme dust outbreak in the Cyprus region based on satellite observations (MODIS, **Moderate Resolution Imaging Spectroradiometer**) aerosol optical thickness AOT, Ångström exponent), surface particle mass ($PM_{10}$) concentrations measured at four sites in Cyprus, visibility observations at three airports in southern Cyprus and corresponding conversion products (particle extinction coefficient, dust mass concentrations), and EARLINET **(European Aerosol Research Lidar Network)** lidar observations of dust vertical layering over Limassol, particle optical properties (backscatter, extinction, lidar ratio, linear depolarization ratio), and derived profiles of dust mass concentrations. Maximum 550 nm AOT exceeded values of 5.0 **according to MODIS** and correspondingly the mass loads were probably $>10$ g/m$^2$ over Larnaca and Limassol during the passage of an extremely dense dust front on 8 September 2015. Hourly mean $PM_{10}$ values were close **to** 8000 $\mu$g/m$^3$, the observed meteorological optical range (visibility) reduced to 300–750 m at Larnaca and Limassol. The visibility observations suggest peak values of the near-surface total-suspended-particle (TSP) extinction coefficients of 6000 Mm$^{-1}$ and thus TSP mass concentrations of 10000 $\mu$g/m$^3$. The Raman/polarization lidar observations mainly indicated a two–layer structure of the dust plumes (reaching to about 4 km height), pointing to at least two different dust source regions. Dust particle extinction coefficients (532 nm) already exceeded 1000 Mm$^{-1}$ and the mass concentrations reached 2000 $\mu$g/m$^3$, respectively, in the elevated dust layers on 7 September, more than 12 hours before the peak dust front on 8 September reached the Limassol lidar station around local noon. Typical Middle East dust lidar ratios around 40 sr were observed in the dense dust plumes. The particle depolarization ratio decreased from around 0.3 in the lofted dense dust layers towards 0.2 at the end of the dust period (11 September) indicating an increasing impact of anthropogenic haze.

# 1 Introduction

On 7–11 September 2015, a record-breaking dust storm hit Cyprus. **According to MODIS (Moderate Resolution Imaging Spectroradiometer), the aerosol optical thickness (AOT) exceeded 5.0 at 550 nm over large parts of the Eastern Mediterranean**. The dense dust clouds originated from Middle East deserts, mainly from northeastern Syria and northern Iraq. Such strong dust storms are rather seldom. **Figure 1 provides an overview of AOT observed with MODIS over Limassol, Cyprus, from 2001–2015. Twelve extreme dust outbreaks reached Limassol in southern Cyprus within the 2001–2015 period. The by far strongest were observed on 1 April 2013 (AOT > 4.0, Saharan dust storm) and 8 September 2015 (AOT > 5.0, Middle East desert dust storm).** Extreme dust events, characterized by an AOT exceeding the climatological mean AOT by four standard deviations, occur, on average, 1-2 per year for a given site in the Mediterranean. The AOT is most frequently lower than 1.5 during these events (Gkikas et al., 2016). An extended aerosol characteristics for the entire Mediterranean region and an extended literature survey is also given by Georgoulias et al. (2016).

**Surprisingly,** dust transport models widely failed to predict this record-breaking dust storm in September 2015 (http://sds-was.aemet.es/forecast-products/dust-forecasts/compared-dust-forecasts). This fact and the enormous dust mass concentrations measured in Cyprus motivated us to investigate the underlying weather conditions that caused this huge dust outbreak. Extreme dust events provide a unique opportunity to learn more about known and established dust mobilizing mechanisms and to identify and explore even new or not well parameterized dust emission processes. The dust storm was obviously linked to an extraordinary weather situation with dust mobilization features on scales which were too small to be resolved by the used global and regional weather and dust transport models. We investigate this extreme dust event in detail by combining the available dust observations in the Cyprus area (presented in this article) with complex atmospheric modeling (presented in the second paper, Solomos et al., Extreme dust storm over Middle-East and the eastern Mediterranean in September 2015: Modeling study with RAMS-ICLAMS, to be submitted to ACP). The occurrence of a haboob in northeastern Syria and northern Iraq was probably responsible for this unique dust outbreak. Haboobs are intense dust storms caused by strong thunderstorm activity, which are associated with density currents (Knippertz et al., 2007; Solomos et al., 2012), strong precipitation and vigorous cold-air downbursts reaching the ground and pushing huge amounts of dust and sand into the air.

The goal of this first article is to provide an overview of the available dust observations in the Cyprus region. We present time series of spaceborne observations (MODIS, Moderate Resolution Imaging Spectroradiometer) of aerosol optical thickness (AOT) for five sites in Cyprus, continuous particle mass concentration measurements (PM$_{10}$, mass concentration of particles with aerodynamic diameter smaller than 10 $\mu$m) at four stations, visibility observations from three airports in Cyprus, and lidar observations, performed at Limassol. We are not aware of any report in the literature in which a severe, record-breaking dust storm has been discussed in so much observational detail. The lidar measurements are especially highlighted in our study. The observed temporally and vertically resolved dust layering structures and the derived profiles of particle extinction coefficient and dust mass concentration provide indispensable information for dust transport simulation studies (presented in the second article). Comparison of modeled and lidar-derived dust profiles are of basic importance in model-based investigations of the

relationship between given meteorological conditions over the dust source regions, dust mobilization, and observed long-range dust transport features (Heinold et al., 2009, 2011; Müller et al., 2009).

Several long-term lidar studies of dust outbreaks towards the Mediterranean are available, however with main focus on Saharan dust outbreaks (e.g. Amiridis et al., 2005; Mona et al., 2006, 2014; Papayannis et al., 2008; Papayannis et al., 2009).
An extreme Saharan dust event with AOT up to 1.5 at 500 nm over southern Spain observed with lidar was discussed by Guerrero-Rascado et al. (2009). **Another lidar study of an exceptionally strong dust outbreak with dust optical depth up to 2.0 towards Portugal was presented by Preißler et al. (2011)**. A first lidar-based long-term study for the Eastern Mediterranean which includes Saharan as well as Middle East desert dust outbreaks has been presented by Nisantzi et al. (2015), based on the Limassol lidar observations.
After the introduction, a brief description of the observation methods, data analysis, and measurement products is given in Sect. 2. The observations are presented and in Sect. 3. Concluding remarks are given in Sect. 4.

## 2   Aerosol instrumentation and observational products

### 2.1   EARLINET lidar profiling of dust optical properties and mass concentration

The lidar observations were conducted by the Cyprus University of Technology (CUT), at Limassol (34.7°N, 33°E, 23 m
above sea level, a.s.l.), Cyprus. The lidar station **is part of** the European Aerosol Research Lidar Network (EARLINET) (Pappalardo et al., 2014) and is equipped with a 532 nm polarization/Raman lidar (nitrogen Raman channel at 607 nm)(Mamouri et al., 2013; Mamouri and Ansmann, 2014; Nisantzi et al., 2015). The EARLINET site is combined with an Aerosol Robotic Network (AERONET) station (Holben et al., 1998; Nisantzi et al., 2014, 2015) and located in the city center of Limassol (see CUT-TEPAK site in the AERONET data base, TEPAK stands for the greek name TEchologiko PAnepistimio Kyprou). Unfor-
tunately, the CUT-TEPAK AERONET photometer was not available from July to October 2015 for calibration reasons.

Details of the lidar data analysis regarding the retrieval of the particle linear depolarization ratio $\delta$, backscatter coefficient $\beta$, extinction coefficient $\sigma$, and extinction-to-backscatter ratio (lidar ratio) $S$, and of the separation of dust backscatter coefficient $\beta_{\mathrm{d}}$ and non-dust backscatter coefficient $\beta_{\mathrm{nd}}$ are given by Tesche et al. (2009a, b), Mamouri et al. (2012, 2013), Mamouri and Ansmann (2014), and Nisantzi et al. (2014, 2015).
The dust mass concentrations $M_{\mathrm{d}}$ is then obtained from the backscatter coefficients $\beta_{\mathrm{d}}$ by means of the equation,

$$M_{\mathrm{d}} = \rho_{\mathrm{d}} c_{\mathrm{v,d}} \beta_{\mathrm{d}} S_{\mathrm{d}} \,, \tag{1}$$

with the dust particle density $\rho_{\mathrm{d}}$, assumed to be $2.6\,\mathrm{g/cm^{-3}}$ (Ansmann et al., 2012), the volume-to-extinction conversion factor $c_{\mathrm{v,d}} = v_{\mathrm{d}}/\sigma_{\mathrm{d}}$ with the dust volume concentration $v_{\mathrm{d}}$ , and the dust lidar ratio $S_{\mathrm{d}}$.

By using a characteristic dust lidar ratio $S_{\mathrm{d}}$ (or even measured ones as during this dust storm), we convert the retrieved pro-
files of the backscatter coefficient $\beta_{\mathrm{d}}$ into respective profiles of dust extinction coefficient $\sigma_{\mathrm{d}}$. We use $S_{\mathrm{d}}$=40 sr for Middle East desert dust (Mamouri et al., 2013). Then, the dust extinction profile is converted into the particle volume and mass concentration profiles $v_{\mathrm{d}}$ and $M_{\mathrm{d}}$, respectively, by using conversion factors from AERONET column observations during pure desert dust

situations. Appropriate conversion factors were derived from extended studies during large dust field campaigns in Morocco, Cabo Verde, and Barbados (Mamouri and Ansmann, 2016). The average conversion factor $c_{v,d}$ is $0.64\pm0.06\times10^{-12}$Mm.

**The uncertainties in all the optical properties, conversion factors and estimated microphysical properties are discussed by Tesche et al. (2009a, b); Gasteiger et al. (2011a, b); Ansmann et al. (2012), and Mamouri and Ansmann (2014).** Relative uncertainties in the dust backscatter and extinction coefficients and lidar ratios are 10–20% at dense dust conditions. Considering in addition a relative uncertainty of 10% in the assumed dust density $\rho_d$ and of about 10% in the conversion factor $c_{v,d}$, we yield an overall relative uncertainty of 20–30% in the estimated dust mass concentrations.

## 2.2 MODIS observations of AOT

MODIS products are used to describe the dust load in the Cyprus region. For five sites we calculated the mean AOT at 550 nm wavelength and mean Ångström exponent (for the 510–670 nm spectral range) from the available set of AOT data within areas with 50 km radius around these cities. **For Limassol, we also calculated the mean AOT for a 25 km radius. Only values that passed a quality check (QAC) are included in the averaging. These are level-2 single pixel AOT(550 nm) measurements with a QAC flag of 3 and $>$0 over land and over the Mediterranean Sea, respectively.** The maximum retrievable AOT is 5.0. **In the data MODIS base (https://ladsweb.nascom.nasa.gov/data/search.html.), all individual validated (pixel) AOT values are set to 5.0 for AOT$\geq$5.0. As will be discussed Sect. 3, this was the case for several stations on 8 September 2015.**

The uncertainty in the retrieved AOT is $0.05\pm0.15\times$ AOT for AOT$\leq$1.0 (Levy et al., 2010, 2013). **Our comparisons of the MODIS products with available AERONET observations at Agia Marina (30 km west of Nicosia) corroborate this uncertainty for AOT$<$1.0. However, MODIS AOT values were systematically larger for AOT$>$2.5. We analyzed Limassol observations (1 April 2013, MODIS AOT of about 4.5, AERONET AOT of 3.2–3.6) and measurements over the AERONET station of the Weizmann Institute, Rehovot, Israel (9-10 September 2015, MODIS AOT of 3.8-3.9, AERONET AOT of 2.4-2.8). Overall, the AOT overestimation was in the range from 0.5–1.5 for AOT(550 nm)$>$2.5.**

## 2.3 PM$_{10}$ observations of the Department of Labour Inspection of Cyprus

Non-validated hourly mean surface observations of PM$_{10}$ concentrations are published by the Air Quality Department of Cyprus (Department of Labour Inspection, DLI, http://www.airquality.dli.mlsi.gov.cy/). We checked the uncertainty in the non-validated hourly values by comparing quality-assured 24-hour PM$_{10}$ values (gravimetric method, European standard, kindly provided by DLI, personal communication, Chrysanthos Savvides) with respective 24-hour mean values calculated from the hourly mean non-validated data. **Uncertainties of the order of $\pm$20–50% must be considered in the discussions of the observations in Sect. 3 as our analysis revealed**.

## 2.4 Visibility observations of the Department of Meteorology of Cyprus

Another way to estimate the dust mass load at ground is based on observations of the so-called meteorological optical range (MOR) $r_{vis}$ (better known as Koschmieder's visibility) (Koschmieder, 1924; Horvath and Noll, 1969; Horvath, 1971). We present visibility time series from three airports in Cyprus (Larnaca, Pafos, and Acrotiri, about 10 km southwest of the Limassol
city center). The data are kindly provided by the Department of Meteorology, Cyprus (DoM, personal communication, Filippos Tymvios). The visibility values are estimated by human observers which are carefully trained after the guidelines of the World Meteorological Organization. The uncertainty of the MOR estimation is of the order of 20–30% for $r_{vis} >$1000 m up to 20 km. For lower MOR, the uncertainty may be considerably higher.

The visibility $r_{vis}$ is linked to the particle extinction coefficient $\sigma$ for 500–550 nm (in the visible wavelength spectrum) by
the relationship (e.g., Horvath and Noll, 1969; Horvath, 1971)

$$\sigma = 3.0/r_{vis} \times 10^6 \tag{2}$$

with $r_{vis}$ in m and $\sigma$ in Mm$^{-1}$. The AOT of 3.0 describes the attenuation of light along the horizontal distance with length $r_{vis}$. Eq. (2) is based on the original Koschmieder formula. Koschmieder (1924) used an AOT of 3.9 which causes an apparent contrast of the object against the bright background of 0.02. The AOT of 3.0 is related to the intuitive concept of visibility
through the contrast threshold taken as 0.05.

**During the strong dust outbreak in September 2015, however, the visibility dropped to values of the order of 300-1000 m, which corresponds to dust extinction coefficients of the order of 3000-10000 Mm$^{-1}$. At these conditions, contributions of marine and anthropogenic particles (including contributions by water uptake) to the total particle extinction coefficient can be neglected. Under clear-air conditions, the extinction coefficient of urban haze at 500–550 nm**
**is 50–150 Mm$^{-1}$ over Limassol (Nisantzi et al., 2015). A typical marine aerosol contribution to particle extinction is 50–100 Mm$^{-1}$ (Mamouri and Ansmann, 2016).**

In order to compare the visibility observations and in situ PM$_{10}$ mass concentrations, we convert the derived particle extinction coefficients $\sigma_d$ into dust mass concentrations $M_d$ by using the relationship (compare Eq. (1))

$$M_d = \rho_d c_{v,d} \sigma_d \tag{3}$$

with the volume-to-extinction dust conversion factor $c_{v,d}$ of $0.64\pm0.06\times10^{-12}$Mm and the dust particle density $\rho_d$ of 2.6 g/cm$^{-3}$, as introduced in Sect. 2.1. The uncertainty mainly depends on the uncertainty in the visibility estimation.

## 3   Results

### 3.1   Dust transport features: Horizontal and vertical dust distribution

Fig. 2 provides an overview about the enormous dust storm in the beginning of September 2015 as seen by MODIS. Optically
dense dust plumes were advected from the east and reached Cyprus on 7 September 2015. Parts of the dust plumes were so

dense that the dark surface of the Mediterranean Sea and eastern and southern parts of the island of Cyprus were no longer visible from space. The highest dust load was observed over Cyprus on 8 September 2015. On this day, the 550 nm AOT clearly exceeded 5 as will be discussed in detail in the next subsection. Unfortunately, lidar observations were not possible on 8 September. We did not switch on the lidar on this day to avoid any potential damage of lidar optics and detection units. The dust amount slowly decreased and showed a second, much weaker maximum on 10–11 September. The Troodos mountains (dark area in southwestern Cyprus) with top heights up to 2000 m were always visible during the dust storm (even on 8 September, AOT>5). This indicates that the thickest dust layers crossed Cyprus at heights below 1500 m height. This conclusion is supported by the lidar observations on 7, 9–11 September.

**To provide a rough idea about the dust source regions and the main airflow during this dust event, Fig. 3 shows six-day backward trajectories for 8 September 2015 (9 UTC). The arrival height of the red trajectory (500 m a.s.l.) is in the lower dust layer which reached to about 1.5 km height according to the Limassol lidar observations on 7 and 9 September. This is also clearly visible in the Nicosia radiosonde profiles of temperature and relative humidity (RH) on 8 September (6 and 12 UTC launches) as will be discussed below. The arrival height of the blue trajectory (2.5 km) is in the upper dust layer (from 1.5–3.8 km as seen by the lidar on 7 and 9 September and indicated by the Nicosia radiosonde profiles on 8 September). The backward trajectories are calculated for a site in the Mediterranean Sea, east of Cyprus (34.7°N, 35°E). Here, the densest dust plumes occurred in the Cyprus area according to Fig. 2 on 8 September 2015.** The HYSPLIT (HYbrid Single-Particle Lagrangian Integrated Trajectory, http://www.arl.noaa.gov/HYSPLIT.php) model was used for this purpose (Stein et al., 2015). Dust from Middle East deserts were transported to the northwest towards northern Iraq and northeastern Syria, and then to the west towards Cyprus.

Figure 4 presents the Limassol lidar observations of the vertical dust layering observed from 7-11 September 2015. **We operated the lidar daily for 3–11 hours up to about local midnight (21 UTC), except on 8 September**. **Dust advection occurred in two to three pronounced, separated dust layers (below about 500-800 m height, and two layers with top heights of 1.5–1.7 km and 3.5-4.2 km height) on 7-9 September**. A first thick dust layer crossed Cyprus in the evening of 7 September between 2 and 3.7 km height. **We speculate that these layer structures also prevailed on 8 September.** This is corroborated by the profiles of temperature and RH measured with radiosondes launched at Nicosia about 60 km northeast of Limassol on 8 September at 6 and 12 UTC.

The peak dust front reached Limassol **at ground** between 8–9 UTC on 8 September **(see photographs below taken briefly after the arrival of the dust front)**. The vertical gradients of temperature and RH were significantly different in the height ranges below and above 1.5 km height. The 12 UTC radiosonde RH profile increased from values of 10–15% at the surface to about 30% at the top of the lower layer in 1.5 km height and indicated well-mixed dust conditions in the Nicosia area. Similarly, the potential temperature was almost height independent and thus also indicated favorable conditions for vertical mixing. In the upper layer from 1.5–3.8 km height, slightly stable conditions were observed.

Figure 5 depicts the dominating two-layer dust structures in terms of dust mass concentration derived from the lidar observations in the evening of 7 September. The values reach 2000 $\mu$g/m$^3$ below 1500 m height and 600 $\mu$g/m$^3$ around 3 km height on 7 September 2015. The two–layer structure of the dust plume is well reflected in the meteorological data measured with

the Nicosia radiosonde on 8 September, 6 UTC, 2–3 hours before the arrival of the main dust front. As mentioned above, the changes in RH and potential temperature with height indicated different air masses and thus different dust source regions above and below about 1500 m height. The meteorological data also indicate that the dust layer was still lofted (base height at around 700 m above ground) in the morning of 8 September, at 6 UTC.

**Similar dust layering structures** were then observed with lidar a day later on 9 September 2015 (see Figure 4), again in consistency with the temperature and humidity profiles of the Nicosia radiosonde (not shown). In the evening of 10 September, another elevated optically dense dust layer crossed the EARLINET lidar station. Finally, on 11 September, a more homogeneous and temporally constant layering was found. The main layer was now below 2 km height. Traces of dust were however detected up to 3–4 km height. On 12 September, **the decrease in the AOT values derived from lidar and MODIS observa-**
**tions indicated the end of the dust episode**.

    In Fig. 6, four photographs taken on 8 September 2015 at local noon (during the passage of the main, rather dense dust front) from the roof of a high building (AERONET station) at Limassol to the south and north are presented. The left photographs show the situation during the phase with the heaviest dust load (8 September, around local noon). These pictures are in strong contrast to the photographs taken one day later, when the dust concentration was still high but the horizontal visibility **increased**
**to more than 10 km.** By careful inspection of the pictures from 8 September (searching for different pronounced buildings and towers) we estimated the horizontal visibility to be 500–600 m. The visibility measurements performed at three airports in Cyprus are discussed in the next subsection. A visibility of 500 m points to dust extinction coefficients of about 6000 $\text{Mm}^{-1}$ according to Eq. (2). If this extremely high extinction coefficient occurred at all heights **up to 800 m or 1.5 km, we end up with AOTs of 4.8 and 9, respectively. These speculative values are in the range of AOTs indicated by the biased MODIS**
**observations.** Such huge dust optical depths indicate column dust loads of 8–15 $\text{g/m}^2$. In the upper layer (above 1.5 km height), the AOT was significantly lower with values around 0.5 or less as the lidar observations on 7 and 9–11 September indicate. This is consistent with the fact that the higher parts of the Troodos mountains remained always visible, even on 8 September in Fig. 2.

    Figure 5 also shows height profiles of the dust outbreaks simulated with the RAMS-ICLAMS model (Regional Atmospheric
Modeling System / Integrated Community Limited Area Modeling System) (Cotton et al., 2003; Solomos et al., 2011). Details to this simulations are given in the follow-up paper (Solomos et al., 2016, in preparation). Dust profiles for arrival times in the evening of 7 September and local noon of 8 September 2015 are shown. The regional model (simulation with 20 km horizontal resolution) clearly underestimates the dust load. As explained in detail by Solomos et al. (paper in preparation) the event seems to be the result of two meteorological processes. A thermal low formed over Syria on 6 September 2015 associated with
strong cloud convection and provided favorable conditions for the generation of a haboob along the borders between Iraq, Iran, Turkey, and Syria on 7 September 2015. Atmospheric density currents evolved and propagated towards the Mediterranean and pushed the pre-existing elevated dust layers towards the Mediterranean Sea. The main reasons that most dust prediction models (including RAMS in regional modeling configuration with too low horizontal resolution to resolve cloud convection processes) did not capture this episode are possibly related to the lack of sufficient physics package to describe the feedback of clouds on

dust mobilization and the lack of sufficient (cloud resolving) model resolution. A detailed discussion is given in the follow-up study (Solomos et al., paper in preparation).

### 3.2 Dust optical properties and mass concentrations: surface and profile observations

Figure 7a shows time series of AOT retrieved from daily MODIS observations for four coastal sites from Risocarpaso at the most eastern tip of Cyprus to Pafos, which is approximately 250 km southwest of Risocarpaso. In addition, the AOT time series for the capital city Nicosia is shown. The mean AOT values for areas with radius of 50 km around these cities are presented. The maximum retrievable AOT is 5.0. As mentioned in Sect. 2.2, many of the individual, validated (pixel) AOT values were set to 5.0 (indicating that the true values were larger). For Fig 7, we used all validated data points in the averaging. **Therefore, all area mean AOT values for 8 September (Julian day 251) exceeding 3.0 have to be interpreted with caution. They include many 5.0-AOT data and are thus biased (towards too low AOT). As outlined in the foregoing section, the uncertainty in the retrievable AOT values is about $0.05 \pm 0.15 \times$AOT for AOT$\leq$1.0. A systematic overestimation of the order of 1.0 must be taken into account for high AOT$\geq$2.5.**

**The AOT clearly exceeded 4.0 in the eastern and southern parts of Cyprus on 8 September 2015, if we assume an AOT overestimation by 1.0. For Limassol, we also calculated the area mean AOT for a 25 km radius around Limassol on 8 September (blue diagonal cross with circle in Fig. 7). The area mean AOT of 4.9 includes many 5.0 values which indicate that the retrieved MODIS AOT clearly exceeded 5.0 over the EARLINET lidar station. It should be mentioned here that the mean AOT values were widely determined by the quality-assured values over the Mediterranean Sea (at all 5.0). Over land, the validated AOT values were significantly lower, mostly 3.0–3.5 on 8 September, 10:30–11:30 UTC. However, over the orographically inhomogeneous terrain, north of the coastal city of Limassol (with surface levels mostly varying between 200 and more than 700 m height a.s.l.), quality-assured data were rare.**

According to MODIS, the AOT ranged from 0.85–1.7 on 9 September 2015, 1.2–2.1 on 10 September, and 1.1–1.4 on 11 September over southern Cyprus (Larnaca, Limassol). Our lidar observations on 7 and 9–10 September reveal that the AOT contribution of the second layer above 1.5 km height was always of the order of 0.5. The AOT was considerably lower at Pafos on 9–10 September, 70 km west of Limassol, with values of 0.4 and 0.3–0.7, respectively. In comparison, our lidar observations (taken about 6–11 hours after the daily MODIS observations) indicate AOTs of 0.5-0.6 on 9 September (MODIS, Limassol, 0.85, Pafos 0.4), 0.7–0.75 on 10 September (MODIS, Limassol 1.2, Pafos, 0.3–0.7), and around 0.85 on 11 September (MODIS, Limassol, 1.1, Pafos, 0.8). On 12 September 2015, all three stations showed significantly reduced dust loads with AOT values from 0.3–0.8 derived from the MODIS observations. In this context it should be mentioned that the relative humidity was always <30%, <40%, <50% within the lowermost one kilometer, up the top of the lower dust layer, and up to the top of the upper dust layer, respectively, on 7–11 September, so that effects of aerosol particle growth by water uptake on the observed AOT values can be neglected. The impact of anthropogenic particles and marine particle may have been of the order of 0.05–0.15 and 0.03–0.05 on the total AOT at 550 nm, respectively **(Nisantzi et al., 2015; Mamouri and Ansmann, 2014)**.

Figure 7b shows that the Ångström exponent (AE), which describes the wavelength dependence of AOT (for the visible wavelength range from 510–670 nm), dropped from typical values of 1.0–1.5 for mixtures of anthropogenic aerosol and marine

particles (and some local dust) to values around 0.3 during the dust period (ignoring the low AE values around 0 on 8 September which are mostly based on biased AOT values).

Figure 7c presents the surface observations of $PM_{10}$ concentrations from 6-14 September 2015. Hourly mean values for five sites across Cyprus are shown. **Uncertainties are of the order of $\pm 50\%$ for the sites of Larnaca, Limassol, and Pafos on 8**
**September 2015, and reduced to about 20% later on (9–11 September) according to the uncertainty analysis described in Sect. 2.3**. The maximum hourly mean dust mass concentration at Limassol was close to 8000 $\mu g/m^3$ on 8 September. **Note that the $PM_{10}$ concentration was only 1300 $\mu g/m^3$ on 1 April 2013 (second largest dust storm during the last 15 years over Limassol in Fig. 1), because Saharan dust layers crossed Cyprus and Saharan dust is mainly transported within lofted layers as our EARLINET lidar observations show.** The quality-assured daily mean values were 2900 $\mu g/m^3$
(Larnaca), 1500 $\mu g/m^3$ (Limassol), and 500 $\mu g/m^3$ (Pafos) on 8 September, 2015.

The $PM_{10}$ observations, which consider particles with diameters $<10$ $\mu m$ only, may have underestimated the total-suspended-particle (TSP) mass concentration. Kandler et al. (2009) showed that the TSP mass concentration can be an order or even two orders of magnitude larger than the respective $PM_{10}$ value during haze periods and when density current-induced dust fronts cross the field site. At Cabo Verde, after long range transport of dust over 1000–3000 km, the TSP-to-$PM_{10}$ particle mass con-
centration ratio was found to be mostly between 1.2–1.5 **(Kandler et al., 2011). The visibility of 500 m according to Fig. 6 is related to peak particle extinction coefficient of 6000 $Mm^{-1}$ and correspondingly to a peak TSP mass concentration of 10000 $\mu g/m^3$. This peak TSP value is about a factor of 1.25-1.3 higher than the in situ measured maximum hourly mean $PM_{10}$ value of around 7600 $\mu g/m^3$, and thus in good agreement with the study of Kandler et al. (2011).**

To further check to what extend the $PM_{10}$ dust observations underestimated the TSP mass concentration during these extreme
dust conditions of 8 September 2015, we analyzed visibility observations at three airports in southern Cyprus. According to Eq. (2) in Sect. 2.4, the visibility is directly related to the particle extinction coefficient, which in turn is highly correlated with the particle volume and mass concentration. The relative uncertainty in the derived mass concentration is estimated to be about 30-40%, provided the visibility is available with an uncertainty of 20-30%.

Figure 8 shows time series of visibility and corresponding extinction coefficient. All three stations show visibilities in the
range from 200-750 m from 5:00 to 20:00 (Larnaca), 6:00–14:00 (Limassol), and 10:00–14:00 (Pafos). The lowest visibilities of 200–300 m values in the Limassol area were observed at Acrotiri airport (about 10 km southwest of the Limassol lidar station) from 8–9 UTC, when the photographs in Figure 6 were taken. The corresponding particle extinction and mass concentration values for Acrotiri are 9000–15000 $Mm^{-1}$ and 15000–25000$\mu g/m^3$, respectively. As mentioned in Sect. 2.4, marine and anthropogenic haze may have contribute to the total aerosol extinction coefficient by about 100–200 $Mm^{-1}$ so that their
contribution to observed extinction values exceeding 2000 or 3000 $Mm^{-1}$ can be ignored in the following discussion and retrievals.

However, if we compare the quality-assured daily mean in-situ measured $PM_{10}$ values for Larnaca (2900 $\mu g/m^3$), Limassol (1500 $\mu g/m^3$), and Pafos (500 $\mu g/m^3$) on 8 September 2015, with the respective daily mean TSP mass concentrations (calculated from MOR values measured every hour), we find visibility-related daily mean TSP mass concentrations of 3600 $\mu g/m^3$
(Larnaca), 2075 $\mu g/m^3$ (Acritori, Limassol), and 1600 $\mu g/m^3$ (Pafos), which are a factor of 2.5 (Larnaca), 2.8 (Limassol), and

6.4 (Pafos) higher than the in-situ measured PM$_{10}$ daily means. These very high (and to our opinion unrealistic) factors of 2.5–6.4 may be caused by a wrong volume-to-extinction conversion factor $c_{v,d} = v_d / \sigma_d$ (a factor of 2 too high) in Eq. (3), or by wrong visibility estimations (leading to a factor of 2 too low $r_{vis}$ values) at these unusual dust conditions. The volume-to-extinction conversion factor is $0.64 \times 10^{-12}$Mm (as discussed in Sect. 2.1). A value around $0.32 \times 10^{-12}$Mm points to conditions with dominating fine-mode dust (Mamouri and Ansmann, 2016). At strong dust outbreak conditions we expect the opposite, namely that coarse-mode dust particles dominate the measured optical effects so that the applied volume-to-extinction conversion factor of $0.64 \times 10^{-12}$Mm is roughly appropriate. **So we conclude that the visibility observations revealed too low visibilities at the unusual conditions on 8 September 2015.**

The next days showed steadily decreasing near-surface dust mass concentrations. The daily mean PM$_{10}$ mass concentration decreased from 2900 $\mu$g/m$^3$ (8 September) to 1000, 500, and 200 $\mu$g/m$^3$ on the following day (9–11 September) at Larnaca, and from 1500 $\mu$g/m$^3$ (8 September) to 500, 200, and 200 $\mu$g/m$^3$ at Limassol on 9–11 September. This steady decrease of the near-surface dust mass concentration was not observed for the total column (see discussion of MODIS and lidar-derived AOTs above) which remained almost constant from 9–11 September.

The highlight of the observations are our lidar observations of the vertical layering of the dust particles. Such profile observations are indispensable in the verification of modeling results and the reliability of model-based dust outbreak studies as a whole. Figure 4 provides an overview of the main dust layering features and indicated mainly a two-layer structure of the advected dust plumes which pointed to two different air mass transport regimes and thus two dust source regions.

In Fig. 9, profiles of particle backscatter and extinction coefficients at 532 nm, the corresponding extinction-to-backscatter ratio (lidar ratio), and the particle linear depolarization ratio at 532 nm for each of the four evenings on 7 and 9–11 September are given. 1-hour to 3-hour mean profiles provide an overview of the main features of the dust optical properties. The backscatter coefficients are obtained with high vertical resolution (signal smoothing window length of 195 m) and show best the layer structures. The profiles of the particle backscatter coefficient and the particle linear depolarization ratio are trustworthy down to 100 m above ground as the comparison with the surface in situ observations (PM$_{10}$ measurements, visibility/extinction observations) corroborate which will be discussed below. The extinction coefficients and corresponding lidar ratios are calculated from smoothed Raman signal profiles (375m smoothing length).

The particle extinction coefficients reached values of 1300 Mm$^{-1}$ in the lower layer and were around 350 Mm$^{-1}$ in the second layer on 7 September. Another pronounced dust front caused extinction coefficients up to 550 Mm$^{-1}$ in an elevated layer between 1000 and 2500 m height on 10 September 2015. The lidar ratios at 532 nm were 35-42 sr in the dust layers on 7 and 10 September, 45-60 sr on 9 September, and 50-60 sr on 11 September. Values of 35–45 sr are typical for desert dust from Middle East dust sources (Mamouri et al., 2013; Nisantzi et al., 2015). Larger lidar ratios on 9 and 11 September indicate a mixture of dust and anthropogenic haze.

The particle linear depolarization ratio assumed typical dust values of 0.25-0.32 (7 and 10 September) in the dense dust layers. These values clearly indicate the dominance of mineral dust in these layers. The decrease towards values of 0.20-0.25 on 9 and 11 September reflects the increasing impact of anthropogenic haze on the optical properties of the advected air masses. The linear depolarization ratio dropped to values clearly below 0.2 in the lowermost 300–500 m thick marine boundary layer

over Limassol and was around 0.1–0.15 at 100 m above ground on 9–10 September. Such low depolarization ratios indicate that anthropogenic pollution contributed to more than 50% to the overall total particle backscattering and extinction coefficients and to 30–50% to the particle mass concentration in the city on 9–10 September. This fact has to be kept in mind when comparing $PM_{10}$ mass concentrations with the mass concentrations derived from the lidar profiles at heights below about 300–500 m.

The backscatter and extinction profiles and the lidar ratio information allow us to estimate the AOT in the lower dust layer (partly from the backscatter coefficients) and to **determine** the AOT at 532 nm in the upper dust layer from the extinction profile. We estimated the extinction values in the vertical range without extinction measurements (in the lowermost about 800 m) by multiplying the backscatter coefficients with a lidar ratio of 50 sr which is higher than a pure-dust lidar ratio and takes the influence of anthropogenic pollution (lidar ratios of 60-80 sr) into account. On 7 September, the 532 nm AOT for the

lower layer (0–1.7 km height) was 1.2, and 0.5 for the upper layer from 1.7–3.5 km according the evening lidar observations. Comparison with MODIS observations before and after noon is not possible on this day because of the rapidly changing dust conditions.

    On 9 September, the 532 nm AOT decreased strongly from the record-breaking values >5.0 on 8 September to values of 0.55 with an AOT around 0.35 for the lowermost 1.2 km height region and 0.2 for the upper dust layer from 1.2–3.0 km height.

In contrast to the evening lidar observations, the morning MODIS data revealed still an AOT of 0.8-1.0 on 9 September.

    Another dense dust outbreak plume reached Cyprus on 10 September. The daytime AOT (MODIS) for Limassol showed a slight increase to 1.0–1.2, the lidar observed an overall AOT of 0.7–0.8 (as three hour average) in the late evening of 10 September 2015. The lower dust layer (up to 1 km height) contributed about 0.2–0.3 and the upper layer (1–3 km) around 0.5 to the total AOT. More vertically homogeneous dust extinction backscatter and extinction profiles were observed on 11 September

with an AOT of around 0.6 for the lower part (0–1.8 km height) of the dust layer and an AOT of about 0.25 for the upper part from 1.8-4.2 km height. MODIS AOT values on 11 September were still around 1.0 (for all stations Larnaca, Limassol, Pafos). Thus a good agreement between MODIS and lidar observations was found for this final dust day.

    We also studied to what extent the lidar backscatter coefficients and the estimated extinction values close to the ground are reliable. Visibility observations yield values for the meteorological optical range of around 10 km in the evening of 9 Septem-

ber, which corresponds to particle extinction coefficients of about 300 Mm$^{-1}$. The lidar measurements indicate backscatter coefficients of 6 Mm$^{-1}$ sr$^{-1}$ close to the surface on 9 September, and thus extinction coefficients of 275 Mm$^{-1}$ (by multiplying the backscatter coefficient with a lidar ratio of 45 sr, representing dust-dominated conditions) to 330 Mm$^{-1}$ (for a lidar ratio of 55 sr, representing a mixture of dust and urban haze).

    An overview of the vertical dust mass distribution, observed in the evenings of 7, 9, 10, and 11 September 2015, is given

in Fig. 10. In Eq. (1), we used a typical Middle East lidar ratio of 40 sr (Mamouri et al., 2013). After the first very dense dust plumes on 7-8 September, another dense dust plume crossed Limassol in the evening of 10 September and the dust mass concentrations was again high with values close to 800 $\mu$g/m$^3$ in the center of the elevated layer from 1000-2500 m height. The two-layer structure vanished on 11 September. Only one layer extending from the surface up to 4.2 km height was observed. In terms of column dust mass concentrations we obtained values of 1.9 g/m$^2$ (for 7 September in Fig. 10), 0.35 g/m$^2$

(9 September), 0.95 g/m$^2$ (10 September), and 0.6 g/m$^2$ (11 September). AOTs of 4.8–9 as estimated for the peak dust front on 8 September indicate peak column dust loads of 8–15 g/m$^2$.

Regarding the quality of the lidar-derived TSP mass concentrations close to the ground, we compared the lidar data with respective PM$_{10}$ observations (mean values for the lidar measurement periods in Fig. 10). The Limassol evening PM$_{10}$ values (considering dust and aerosol pollution) were 55 $\mu$g/m$^3$ (7 September), 120 $\mu$g/m$^3$ (9 September), 125 $\mu$g/m$^3$ (10 September), and 165 $\mu$g/m$^3$ (11 September). The respective lidar-derived total aerosol mass concentrations were 65 $\mu$g/m$^3$ (7 September), 180 $\mu$g/m$^3$ (9 September), 125 $\mu$g/m$^3$ (10 September), and 290 $\mu$g/m$^3$ (11 September). The uncertainties are roughly 30% for the lidar mass values and 50% for hourly-mean PM$_{10}$ values. Again, good agreement is obtained **taking** the uncertainties in the derived values into account. Horizontally inhomogeneous downward mixing of dust and horizontal inhomogeneities in the mixture of dust and urban pollution may have also contributed to the differences. Note, that Fig. 10 only shows the dust-related mass concentrations. The contribution of urban and marine aerosol to the TSP mass concentration was of the order of 20–30 $\mu$g/m$^3$ (7 and 10 September) and 40–50 $\mu$g/m$^3$ (9 and 11 September 2015).

## 4 Conclusions

A record-breaking dust storm over the Eastern Mediterranean in September 2015 has been documented and discussed based on satellite, lidar, and in situ aerosol observations in the Cyprus area. We were able to provide a consistent picture of this dust event in terms of a variety of optical and microphysical, and dust layering properties obtained by means of very different in situ and remote sensing observational techniques and retrieval approaches. The presented documentation of an extreme dust storm is a valuable contribution to the literature dealing with long-range transport of dust, forecasting of dust outbreaks, and the research on the relationship between meteorological conditions and dust emission **intensity.**

Such unique events may take place once in a decade or even less frequently and are thus obviously linked to unique mete-orological **conditions**. The documentation of extremely seldom dust storms with vertical, horizontal and temporal resolution (in this article) in combination with advanced atmospheric modeling covering cloud evolution, development of thunderstorm, density currents, dust mobilization and dust transport (in the follow-up article) will certainly lead to an improved understanding of the evolution of dust storms at extreme meteorological conditions. The modeling studies will further reveal what kind of modeling infrastructure is required to resolve even small-scale hot spots of dust mobilization phenomena in order to improve dust forecasting in general.

Another concluding remark deals with the need of a dust lidar network around the main desert areas, e.g. in the Europe-Africa-Asia region from the Sahara, over the Middle East deserts to the desert regions in central, southern and eastern Asia. Continuously operated lidars would be an ideal supplement to dust forecast dust model efforts with the potential goal to assimilate the lidar products into the forecast models. As demonstrated in this article, modern polarization lidars allow us to separate dust and non dust optical properties and to quantify the dust-related particle extinction coefficient and mass concentration in the vertical profile with an uncertainty of 20-30%.

*Acknowledgements.* The authors thank the Eratosthenes Research Center of CUT for support. R.-E. M. would like to thank CUT's library for the financial support within Cyprus University of Technology Open Access Author Fund. The authors acknowledge support through the following projects and research programs: ACTRIS Research Infrastructure (EU H2020-R&I) under grant agreement no. 654169, BE-YOND (Building Capacity for a Centre of Excellence for EO-based monitoring of Natural Disasters, FP7-REGPOT-2012-2013-1) under

5 grant agreement no. 316 210, BACCHUS (impact of Biogenic vs. Anthropogenic emissions on Clouds and Climate: towards a Holistic UnderStanding, EU FP7-ENV-2013) under grant agreement project number 603445, and GEO-CRADLE (EU H2020 R&I) under grant agreement No 690133. The authors are very thankful to the Air Quality Department (Department of Labour Inspection, DLI) for establishing and maintaining the air quality stations of Republic of Cyprus, and Dr Chrysanthos Savvidis (DLI) for providing quality–assured $PM_{10}$ daily means. We further thank Dr Filippos Tymvios from the Department of Meteorology (DoM) of Cyprus for the visibility observations. The

10 authors are grateful to AERONET for high-quality sun/sky photometer measurements. We gratefully acknowledge the NOAA Air Resources Laboratory (ARL) for the provision of the HYSPLIT transport and dispersion model as well for the provision of Global Data Assimilation System (GDAS) data used in this publication. The Terra/MODIS Aerosol Daily datasets were acquired from the Level-1 & Atmosphere Archive and Distribution System (LAADS) Distributed Active Archive Center (DAAC), located in the Goddard Space Flight Center in Greenbelt, Maryland (https://ladsweb.nascom.nasa.gov/). We acknowledge the use of data products or imagery from the Land, Atmosphere

15 Near real-time Capability for EOS (LANCE) system operated by the NASA/GSFC/Earth Science Data and Information System (ESDIS) with funding provided by NASA/HQ.

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

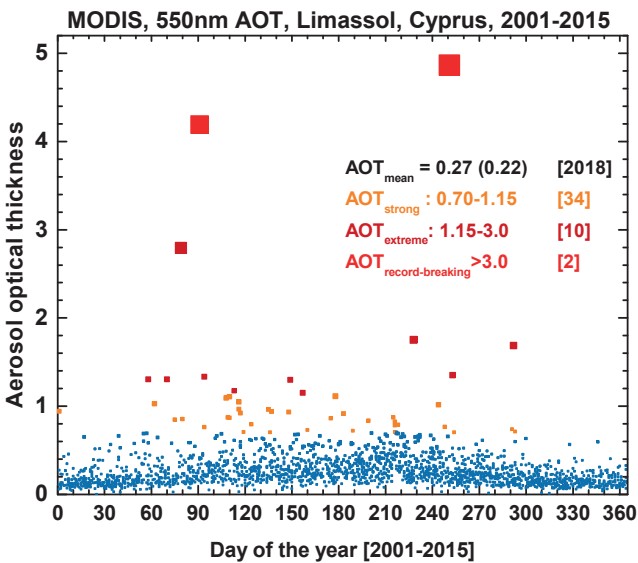

**Figure 1. Seasonal distribution of 550 nm AOT at Limassol based on 15 years of MODIS observations (http://lance-modis.eosdis.nasa.gov/, terra MODIS, 2001–2002, aqua MODIS, 2003–2015). The mean AOT for an area centered at Limassol with 25 km radius is shown. Numbers (in black) indicate total number of considered daily observations (in brackets), the 15-year mean AOT (M), the standard deviation (SD, in parentheses), strong dust events (AOT from M+2SD to M+4SD), extreme dust events (AOT from M+4SD to 3.0), and record-breaking dust events (AOT>3). MODIS collection-6 data were used for the years 2005-2015 and collection-51 data for 2001-2004. More details to the MODIS data including AOT retrieval uncertainties are given in section 2.2.**

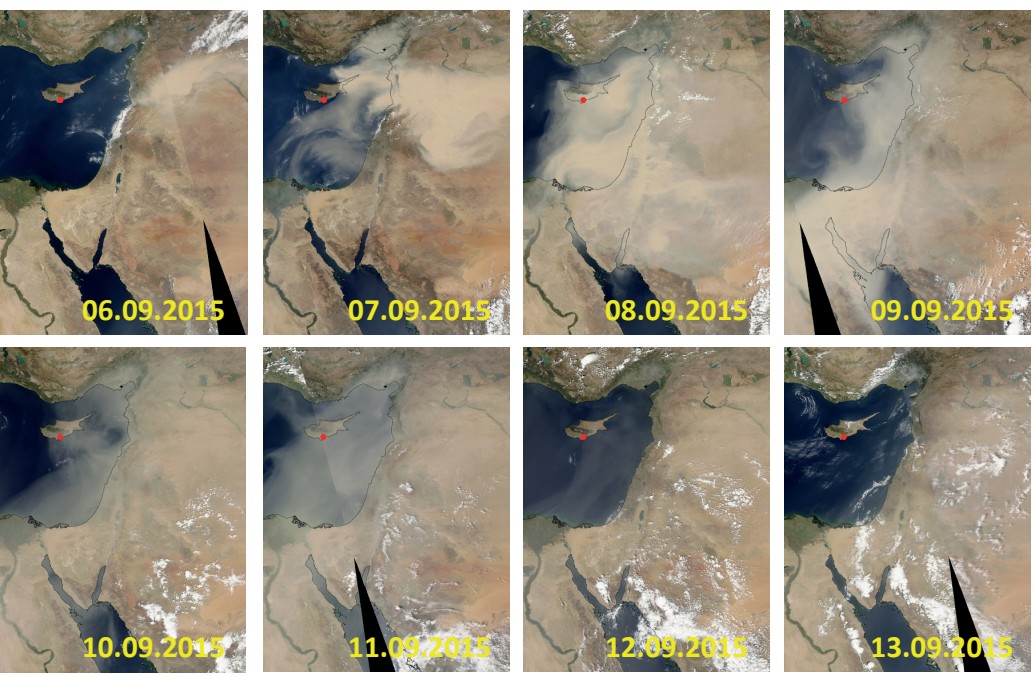

**Figure 2.** Dust outbreak towards Cyprus in September 2015 as seen from space (aqua-MODIS, 10:30-11:30 UTC overpasses, 13:30–14:30 EEST, Eastern European Summer Time). **Red points indicate Limassol.**

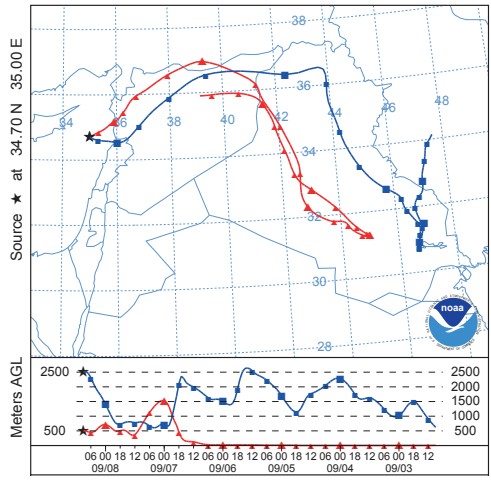

**Figure 3.** Six–day HYSPLIT backward trajectories (http://www.arl.noaa.gov/HYSPLIT.php) arriving in the Cyprus region at 35°E (about 160 km east of the Limassol lidar station) at 500 m (red, lower dust layer) and 2500 m height (blue, upper dust layer) on 8 September 2015, 09:00 UTC (12:00 EEST).

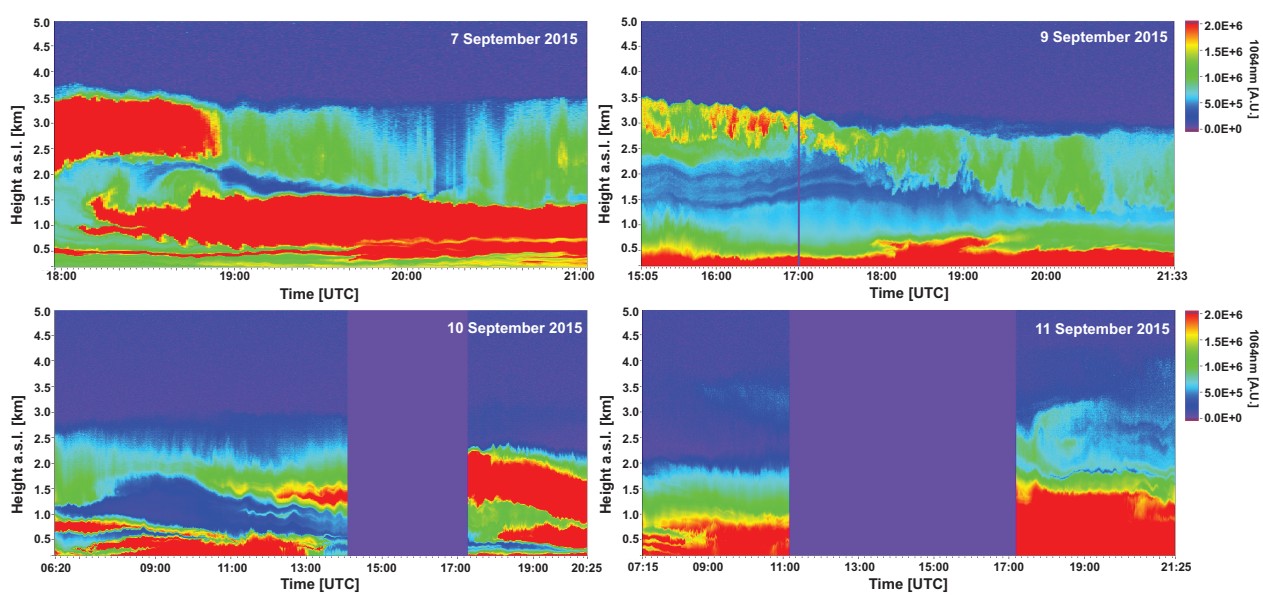

**Figure 4.** Desert dust layers observed with lidar over the EARLINET station of Limassol, Cyprus, on 7, 9, 10, and 11 September 2015. Range–corrected 1064 nm backscatter signals (in arbitrary units, A. U.) are shown. On 7-10 September, a two-layer structure dominated with dust layers below about 1-1.7 km height and another layer reaching to 2.5-3.7 km height. Local time (EEST) is time in UTC plus 3 hours.

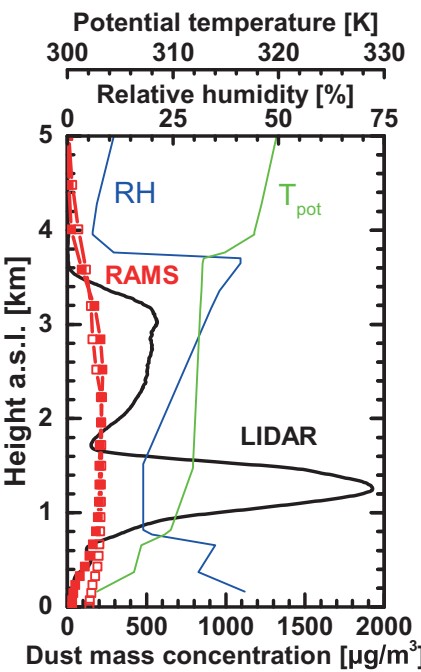

**Figure 5.** Mean dust mass concentration observed with lidar (thick solid black line) at Limassol on 7 September, **19:00–20:00 UTC**, and dust profiles simulated with RAMS (normal run with 20 km horizontal resolution) for Limassol, on 7 September, 18:00 UTC (closed red squares), and 8 September, 9:00 UTC (open squares). Radiosonde observation (launched at the radiosonde station at Athalassa near Nicosia on 8 September 6:00 UTC) of height profiles of potential temperature ($T_{pot}$, thin green curve) and relative humidity (RH, thin blue curve) are in good agreement with the two-layer dust structures observed about 12 hours earlier. The lofted dust layer from 1.7-3.6 km height was well mixed.

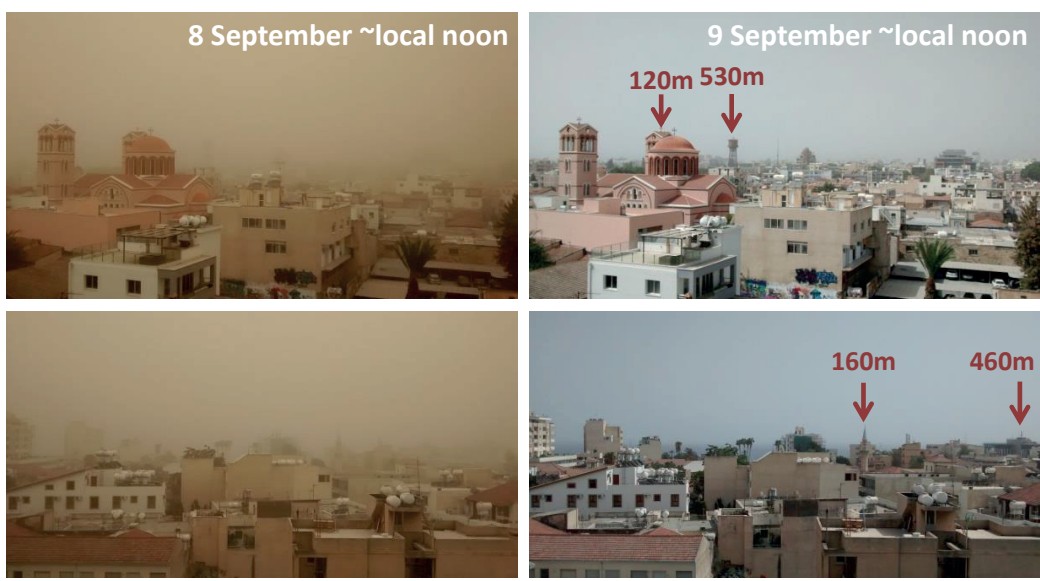

**Figure 6.** Photographs taken **from** the roof of a high building (CUT-TEPAK AERONET site) in the city center of Limassol to the north (top) and south (bottom) on 8 September 2015, 8:20-8:30 UTC (left) and on 9 September 2015 (right), again around local noon. The meteorological optical range (or horizontal visibility) was about 500 m on 8 September and $\geq$10 km on 9 September 2015. Distances to several towers from the AERONET station are indicated.

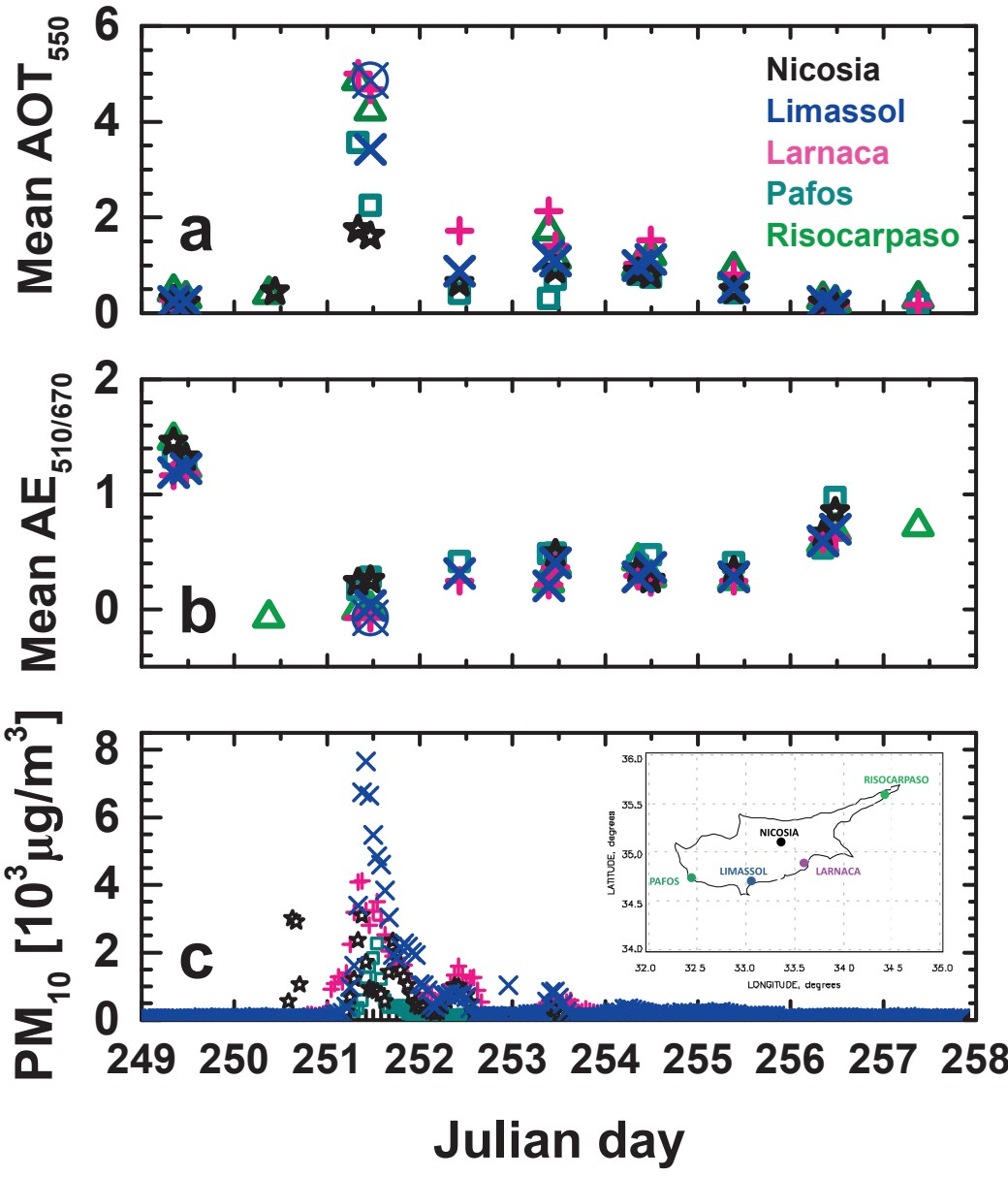

**Figure 7.** (a) MODIS-derived mean 550 nm aerosol optical thickness (AOT) for five sites in Cyprus for the period from 6-14 September 2015 (Nicosia, stars, Limassol, diagonal crosses, Larnaca, crosses, Pafos, squares, Risocarpaso, triangles, aqua-MODIS, 10:30-11:30 UTC, and terra-MODIS, 8:00–9:00 UTC overpasses), (b) MODIS-derived Ångström exponent (for the 510–670 nm wavelength range), and (c) hourly mean $PM_{10}$ particle mass concentrations measured at four stations in Cyprus (Nicosia, Limassol, Pafos, Larnaca). **The AOTs are determined from all MODIS values within areas with 50 km radius around a given city, except the AOT indicated by the blue diagonal cross with circle (Limassol, 25 km radius, day 251).**

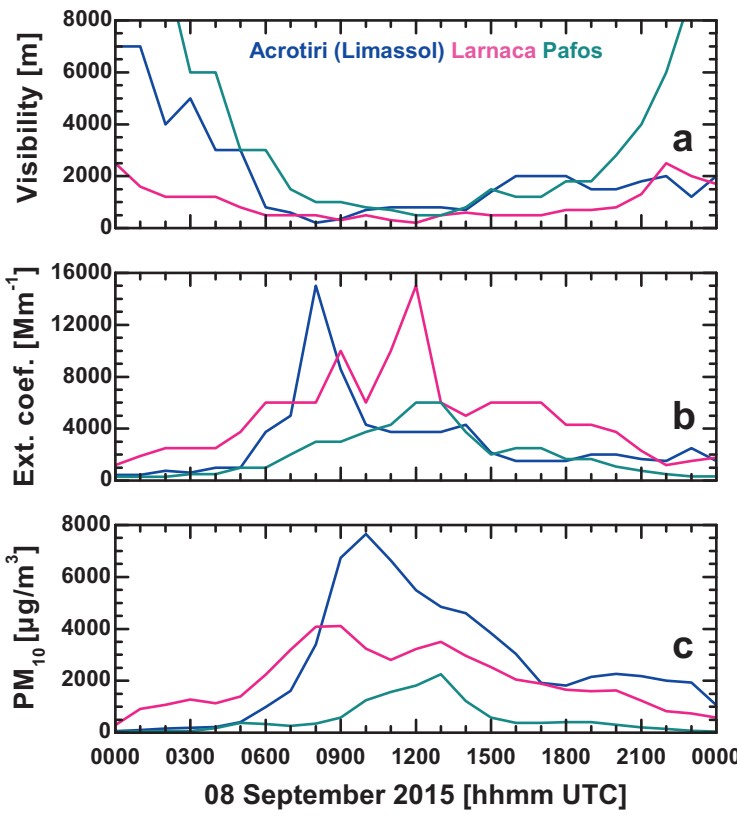

**Figure 8.** Visibility ($r_{vis}$ in Eq. (2)) measured at three airports in southern Cyprus (see map in Fig. 7c) on 8 September 2015, (b) corresponding dust extinction coefficient (by using Eq. (2)), and (c) $PM_{10}$ concentrations (same as shown in Fig. 7c). Relative uncertainties in all parameters are of the order of 50%. Dust extinction coefficients of 4000–8000 $Mm^{-1}$ indicate dust mass concentrations of 6600–13300 $\mu g/m^3$.

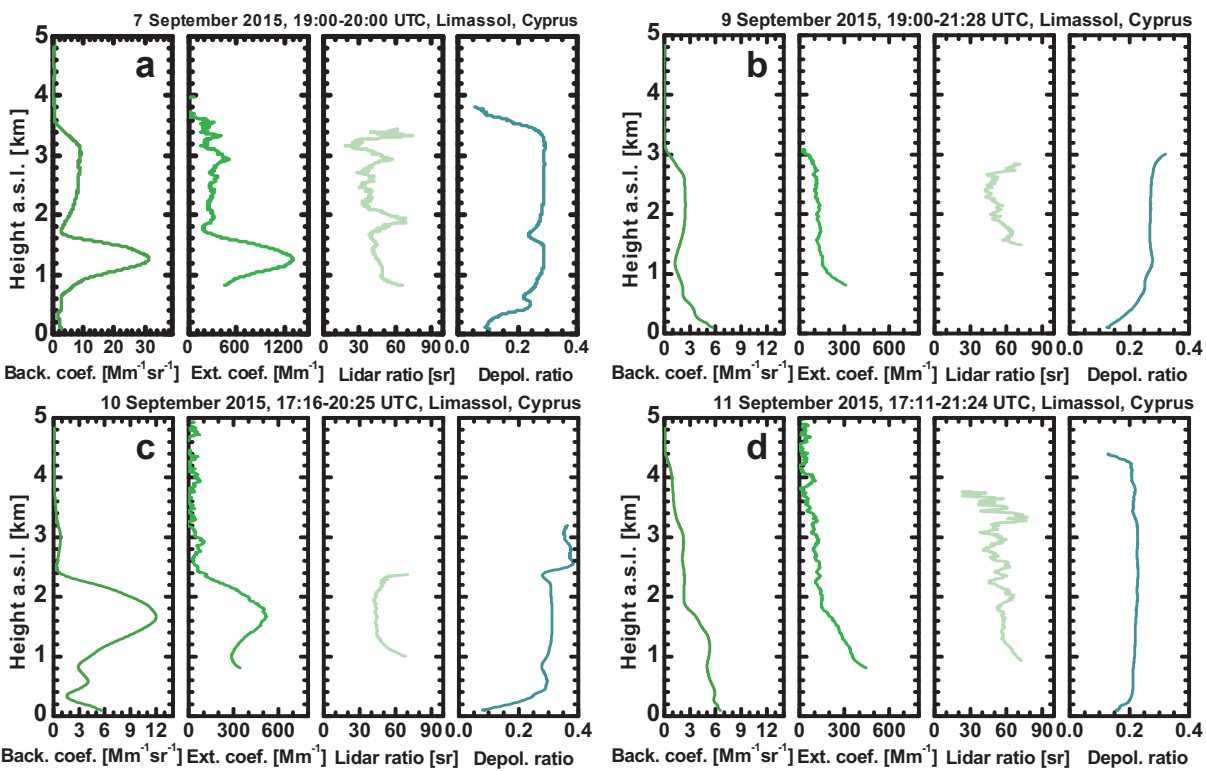

**Figure 9.** Mean vertical profiles of the 532 nm particle backscatter coefficient, extinction coefficient, lidar ratio, and particle linear depolarization ratio for the observational periods given on top of the panels on 7–11 September 2015. The Raman lidar method is applied. Retrieval uncertainties are of the order of 10% (backscatter coefficient, depolarization ratio) and of 20% (extinction coefficient, lidar ratio) at dense dust conditions.

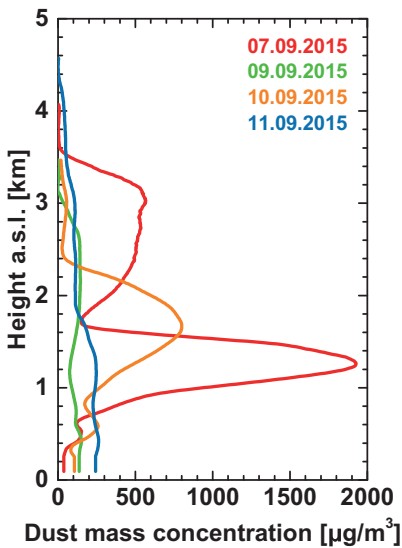

**Figure 10.** Lidar-derived mean dust mass concentrations for the evening periods (see Fig. 9) of 7 September (19:00–20:00 UTC), 9 September (19:00–21:28 UTC), 10 September (17:16–20:25 UTC), and 11 September (17:11–21:25 UTC). The overall uncertainty in the retrieval of the dust mass concentration is of the order of 25%.