# Peer review of "Extreme dust storm over the eastern Mediterranean in September 2015: Satellite, lidar, and surface observations in the Cyprus region"

_Atmospheric Chemistry and Physics, 2016_

## Referee Comment (RC1) · Anonymous Referee #2 · 16 May 2016

[11pt]article

[english]babel

Extreme dust storm over the eastern Mediterranean in September 2015: Lidar vertical profiling of desert dust at Limassol, Cyprus

**Review of manuscript number: acp-2016-354**

May 16, 2016

**1  General comments**

This article describes an exceptional dust event observed in Cyprus. The authors combined remote sensing from ground and space with ground-based in-situ aerosol measurements and models to give a comprehensive overview of the dust plume. The paper is rather descriptive, but the described methods are sound and the data set is unique. Therefore, I would recommend the paper to be published in ACP. However, there are some fundamental points that need to be addressed before publication. My specific comments and technical corrections are given below.

**2  Specific comments**

Both, introduction and conclusions, should be reworked. The introduction is giving results described later in the text, which is not appropriate. Besides, in a rather short and straightforward paper like this, a detailed description of the structure of the paper
is not necessary. I would suggest to present a stronger, more concise motivation. Also, the state of knowledge on Middle East dust is not discussed and should be included.

The conclusions are rather a summary of the results. One of the review criteria for publication in ACP is the following: "Are substantial conclusions reached?" Please consider this point, which is the main weakness of this manuscript in my opinion.

What I'm also missing are information on the difference between local time and UTC. Besides, it would be very beneficial to include more details on the model forecast. The failure of the model is mentioned a few times, but what was actually forecasted?

Some more detailed comments:

**page 1, line 3** Please include a better discussion about the models failing to predict the event in the main text. You highlight it in the abstract and it is repeated in the main text, but it is not shown.

**introduction** The first paragraph of the introduction is a summary of results. Please remove. Also I'm missing some lines motivating this study. It is remarkable, but why is it important to study such cases in detail? Please also discuss literature on Middle East dust.

**page 2, line 2** replace "accurately determined" by "estimated"; I don't think an estimation "around 500 m, but clearly below 750 m" should be called accurate.

**page 2, line 4** Also here I have some concerns about the "good accuracy". You don't give uncertainties in your estimate, but a range of values. Which is ok, but a range of 3000 $\mu$g/m$^3$ should not be called accurate. You present a simple and very nice method to estimate the dust load, but the emphasis here is on "estimate". It is not a precise measurement and should not be presented as such.

**page 2, line 10** Why were data at 1064 nm not used for the dust characterisation, for example to calculate the Ångström exponent (only for time-height plots, figure 5)?

| | |
|---|---|
| **page 2, line 12** | Which of the cited publications describe the set-up used for this work? |
| **page 2, line 13** | Which of the cited publications describe the data analysis used in this study? It would be helpful to at least give the main points of the analysis and specify which reference describes what. |
| **page 3, lines 17-18** | Could you please discuss the influence of humidity and consequent hygroscopic growth of the dust particles on the dust mass concentration? |
| **page 4, line 12** | I would interpret "morning" and "noon" as indicators of local time. But either way, 14:20 is afternoon. It's a small detail, but can be confusing if you are not familiar with the time zone of Cyprus. Rather be specific. Local noon is 12 local time. |
| **figure 6, caption** | How did you obtain the uncertainties? |
| **page 5, line 10** | A "front" is a distinct line (or surface); the front of the plume. The passage of it should not take three days. I suggest you rephrase by replacing "dust front" by "dust plume" or similar. |
| **page 5, line 13** | Who defined this threshold? And why? |
| **page 5, line 20** | Predicted by which model(s)? What did it/they predict for $35°$? And for $33°$? |
| **conclusion** | It is a summary rather than conclusions. Try to make a stronger case for your findings. What is the contribution of this work to existing knowledge? And how does it link back to your motivation? |

**3  Technical corrections**

| | |
|---|---|
| **page 1, line 20** | replace "re-analyze" by "re-analyzing" |
| **page 1, line 24** | replace "imaginary" by "imagery" |
| **instrumentation** | I suggest to rename this section to "Instrumentation and methodology". |
| **page 2, line 18** | replace "imaginary" by "imagery" |

| **page 2, line 20** | replace "dust plumes were partly" by "parts of the dust plumes were" |
|---|---|
| **figure 2** | Why are there two dots at the same location each day? Are those from MODIS on Aqua and on Terra, respectively? Please specify this in the text or caption. |
| **figure 5, caption** | In my opinion the following part of the caption should be included in the main text body as part of the discussion of the figure: "The signals backscattered by dust in the elevated layers above 1000-1500 m height are partly strongly attenuated by the desert particles occurring below 1500 m. As a consequence, the elevated layers are mostly given in blue and green instead of red (as it would be the case after the correction of the attenuation effect)." |
| **page 3, line 31** | Please include lines 3 and 4 from page 4 (starting with "Unfortunately, ..." ending with "... detection units.") after the sentence ending with "... the highest dust load.". |
| **page 3, line 31** | I don't think it is obvious, as you don't show lidar profiles from that day. You could rephrase this sentence, e.g. "Judging from the MODIS observations in Fig. 1., the rather thick dust layer reached 1000-1500 m height on 8 September." |
| **page 4, line 12** | replace "session" by "sessions" |
| **page 4, line 19** | replace "is" by "was" |
| **figure 6** | Please change the labelling on the x-axis of the depolarisation ratio plot. It is hard to tell the numbers apart. You could leave major ticks at 0.2 and 0.4 (with labels) and use minor ticks at 0.1 and 0.3 (without labels). |
| **figure 6, caption** | The following parts should be included in the main text body rather than the caption: "The Raman lidar method is applied.", and " Retrieval uncertainties are of the order of 10% (backscatter coefficient, depolarization ratio), 25% (extinction coefficient), and 30% (lidar ratio)." |
| **figure 6, caption** | Add reference to "Raman lidar method". [now in text body] |

| **page 5, line 10** | replace "extrem" by "extreme" |
| **page 5, line 15** | Please specify length of trajectories in days in the text. It is in my opinion not enough to include it in the caption of figure 8. |
| **page 5, line 19** | replace "dust advection" by "air mass transport" |
| **figure 7, caption** | Please move the following part of the caption to the main text body: "A dust particle mass density of 2.6 g/cm$^3$ is assumed in the retrieval. The overall uncertainty is 30% and mainly caused by the uncertainty in the dust volume-to-extinction ratio (extinction-to-volume conversion factor) assumed to be $0.8 * 10^{-6}$ m." |
| **figure 7, caption** | Add references for the mass density, conversion factor and uncertainty estimation. [now in text body] |

---

## Referee Comment (RC2) · Anonymous Referee #4 · 31 May 2016

This brief study presents lidar measurements acquired during an extreme dust storm which swept across the Middle East in September 2015. The lidar observations are high quality and are fascinating. I have no particular issues with the data processing etc... However, I think that the scientific content of the paper is weak. I do not recommend publication in ACP, for the reasons detailed below.

I believe he paper would be a better match for AMT.

General Comments

1. In particular, I am missing an analysis of the synoptic situation leading to the dust storm, and how this situation evolved and led to the demise of the dust storm as ob-

served with MODIS and the lidar in Limasol. I am also missing a detailed analysis of the processes leading to the dust emission and transport, as well as the activated source regions. In several instances, the authors state that dust transport models failed to predict this record dust event, without giving precision on the models, their resolution, etc... An analysis of such processes would be extremely useful to understand why the transport models did fail, if they did.

2. I also find the interpretation of the HYSPLIT back trajectories to lack insightful analysis. The authors find them to be in contradiction with their dust observations, and ascribe the differences to erroneous meteorological fields provided by global scale models. This may well be the case but the main issue is that HYSPLIT cannot not deal with turbulence and transport in turbulent planetary boundary layers (PBLs). However, this is not verified in the present study and one wonders what is the worth of including these back trajectories in the analysis (?).

3. Emphasis is put in the abstract on the supposedly large AOD and mass concentration during the event. However, these values are not observed on 8 September (the day when the storm was most intense), and are only speculative, extrapolated from indirect measurements, or based on assumptions. They can be discussed in the text, but should not be emphasized that much in the Abstract and in the conclusion (where they are presented as results, see p5, l 28: "Dust AOT values of the order of 6-10 occurred over Cyprus [...]"). Furthermore, the very large AOD values (6-10) inferred on that day from pseudo-lidar measurements and yet MODIS retrievals on the same day are thought to be biased (p2, l30).

4. The 2-layer structure observed by lidar on 7 September suggests different emission source regions and transport regimes from Middle East, as is the case in other parts of the world, like the Sahara. This structure likely relates to differential advection and PBL dynamics over the emission regions. Knowledge of these processes would be extremely useful for a comprehensive analysis of the back trajectories.

Minor comments

P1, l 14: where does the number 1000 $\mu$g m-3 come from? P1, l 17: which models do you refer to? P1, l 24: imagery

P2, l20: [. . .] we in some parts so dense [. . .] P2, l 24-25: yet you show later on that dust plumes were observed by lidar above that height. . . Does this mean that the dust plume transported at higher altitude was less optically thick? P2, l 30: why are they biased?

P3, l 24-26: how do you know particles bigger that 10 $\mu$m are transported to Cyprus? Furthermore, what is the influence of marine aerosols on these measurements? Would not they influence the surface mass concentration measurements (Limasol is close to the sea)? P3, section 3.3: in this section, nothing is said about the high backscatter values observed in the lower layer (island or marine PBL?). Do not you expect dust to also be present in the lower layers as the result of entrainment at the top of the PBL? P3, l30: regarding the 2 layer structure: you do not have lidar data, but do you have met soundings to show the suggested layering?

---

## Referee Comment (RC3) · Anonymous Referee #1 · 1 Jun 2016

The paper describes an exceptional dust storm over Cyprus. Using lidar data, visibility studies, ground based PM10 measurements and several assumptions it is tried to characterize the event. This should help to improve transport models which – according to the authors – totally failed to predict this outbreak. The paper is clearly structured and easy to understand. It focusses on nothing but the dust storm – thus it is quite short. I did not find any errors.

I think that the paper can in principle contribute to two aspects: validation of transport models or (optical) characterization of desert dust. In both cases the current version must be improved significantly.

According to the manuscript the main motivation is to provide information for the im-

provement of transport models. However, no strategy how to reach this goal is detailed, not even outlined. Which models are meant (in the introduction the authors mention 'state-of-the-art dust transport models' [plural], but only HYSPLIT is mentioned later)? Are the depolarization ratio and the lidar ratio useful parameters to improve a transport model? How can these parameters actually contribute to this goal? Would it be sufficient to determine only the backscatter coefficient and the PM10-concentration to check any improvement of the model? According to Section 3.6 wrong meteorological fields were primarily responsible for the bad forecast (not the numerical description of the microphysics of the model). Is it necessary to estimate the vertical profile of 8. September (this is really vague!) for this purpose or would an agreement for the other days be sufficient to check the improvement of the model? From this point of view I recommend to combine this paper with the 'follow-up'-paper mentioned on page 5 line 25. Then it would be possible to clearly demonstrate the role of the data for the improvement of the model, and to show whether it was successful or not.

The other aspect – I had expected this when I saw the paper the first time – is to study optical properties of desert dust. Though there are already many similar papers on dust of different deserts and different transport paths (mixing, aging), this paper could potentially be a useful contribution: it comprises the 'standard output' of advanced lidar systems (lidar ratio, depolarization ratio, wavelength-dependent extinction and backscatter coefficients), provided that the accuracy of these parameters will be added. If the source region can clearly be identified this study can be a contribution towards a climatology of optical properties of different deserts.

A combination of both aspects would certainly be the most attractive solution and together with the improvement of the transport model this paper could be a really interesting scientific study (different from many previous 'dust-papers') and would fit to ACP.

By the way: it is surprising that more than 80% of the references are from the authors themselves. I am sure that there are more publications on these topics.

A few minor comments:

- 1-12: 'hit' change to 'hits'

- 2-14: What is CUT-TEPAK?

- 3-10: 'observations from pilots': where are these information actually coming from? What exactly do they provide?

- 3-16: Is this from the Koschmider's formula?

- 3-22: 'are not validated': What does this mean? What errors can be expected? Would it have been possible to validated these data (under which conditions)?

- 4-6: 'AOT close to 1': where is this estimate coming from?

- 5-8: 'The technique applied ... is described': This should be briefly outlined here so that the reader immediately can understand which measurements and which assumptions are used.

- 5-10: typo: 'extreme'

- Fig. 7: I would expect that the curves in Fig. 7 (dust concentration) are proportional to the backscatter profiles shown in Fig. 6. At least for 10. September this seems not to be the case. This emphasizes the need to better explain the way how the mass concentration is determined (see previous comment).

[Figure]

---

## Author Comment (AC1) · 1 Jun 2016

We would like to thank already now the three reviewers for their valuable comments (although one month is left for further discussions). However, we feel we should immediately respond to the concluding recommendation of reviewer 4, who stated: I do not recommend publication in ACP.

An extended answer to all points of all reviews (very good and constructive suggestions!) will follow later (after closing the open discussion in July 2016).

For us, there is no doubt that the contribution (as it is) is a significant contribution to atmospheric science. We are convinced that the work is clearly worthwhile to be

published, and that ACP is the right (and appropriate) journal for such kind of reporting papers.

What is the basis for our argumentation?

We report and document a unique, extreme (record) dust storm over Cyprus. To our best knowledge such a case has never been reported (in such quantitative detail) in the literature. We think that alone this fact, that we report a unique (record) dust outbreak, is already sufficient to justify publication. We think, there must always be room in scientific journals just for 'breaking' NEWS (unique events). Such papers are needed to stimulate new science directions, alternative research paths, new proposals, especially when dust prediction models failed to predict such an extreme dust outbreak.

We agree that we have to better combine the observations with modelling efforts. Therefore we want to state here that we planned from the beginning to have two papers. The first paper deals with the observations and carefully elaborated measurement results and the second paper will concentrate on the modelling results (a regional atmospheric model is used) and will discuss the reasons for the bad predictions as well as potential solutions how to avoid such modelling situations in future. So, as a consequence of the reviews, we will show some modelling results already in the paper under review. But we do not like the idea to combine all the observations with all the (already available) complex modelling results in one paper. Such a paper would be simply too long.
* * *

---

## Referee Comment (RC4) · Anonymous Referee #3 · 8 Jun 2016

Review of the paper "Extreme dust storm over the eastern Mediterranean in September 2015: Lidar vertical profiling of desert dust at Limassol, Cyprus" by Mamouri et al. submitted to Atmospheric Chemistry and Physics.

This short paper describes the case of an extreme dust plume occurred over Eastern Mediterranean lasting for few days and observed with ground based measurements at Cyprus. In addition satellite products from MODIS are used. The paper is within the scopes of Atmospheric Chemistry and Physics as it presents a rare (may be unique) dust event. However, in its current version needs major revision before it can be accepted for publication. The two main reasons is the lack of uncertainties throughout the paper and the presentation of speculative results (although not necessarily wrong)

in the aim to characterize the event especially for the 8th September when lidar data were not available. For more details see the comments below.

Major comments

1) Cyprus has four WMO stations (https://www.wmo.int/cpdb/dashboard/index/countryCode:CYP). Why you do not present visibility results from them? At least 3 (Akrotiri, Larnaka and Paphos) of them take visibility observations, with Akrotiri being next to Limassol (∼25 km). The data should be available at least through the national meteorological service of Cyprus. Although, the photographs in Figure 3 are indicative of the low visibility occurred during the noon of 8th and the contrast with 9th September, still they do not follow exactly the WMO guidelines especially when visibility is poor, as established in WMO Guide to Meteorological Instruments and Methods of Observation (http://library.wmo.int/pmb_ged/wmo_8_en-2012.pdf). The visibility results will permit also an assessment of the variability of the dust extinction-to-mass conversion factor given the observations of PM10 for the 3 sites providing both types of observations.

2) For a short paper like this presenting an extreme event, what I was hoping to find was uncertainties to all the observations and of course the factors used from previous studies. Just providing mean values is not enough in order to establish the importance and the uniqueness of this event, especially for the values mentioned in the abstract and conclusions. Although, the authors provide estimations in the legends of Figures 6 and 7, it would be better to introduce them in the manuscript and the figures. While there are not uncertainties presented regarding the results of 8th September.

3) In the abstract and the conclusions you present mass concentrations values estimated from visibility by using a typical dust extinction-to-mass conversion factor. Also, the AOD values of 6-10 are based on assumptions about the vertical distribution. Although, in both cases the values are logical (according to the analysis presented in Section 3), simply they are not measured, so I strongly suggest to present in the abstract and the conclusions only the observations by adding the phrase "with possibly

higher values occurred the 8th September" or something similar. Especially, in the case of mass concentration there are measurements and PM10 (or PM2.5) is what you find in the literature. For this reason is more appropriate to present the PM10 measurements in the abstract/conclusions.

4) The Introduction needs rewriting, especially paragraphs 1 and 3. What's the point to present the results of the study already in the Introduction section in a short paper like this? On the other hand, there is no reference at all about climatological and extreme event studies for the region. At least, there are studies covering the Eastern Mediterranean dealing with AOD, lidar measurements and PM10. Example of relevant papers (certainly a non exhaustive list) can be found below:

Basart, S., Pérez, C., Cuevas, E., Baldasano, J. M., and Gobbi, G. P.: Aerosol characterization in Northern Africa, Northeastern Atlantic, Mediterranean Basin and Middle East from direct-sun AERONET observations, Atmos. Chem. Phys., 9, 8265-8282, doi:10.5194/acp-9-8265-2009, 2009.

Gkikas, A., Hatzianastassiou, N., and Mihalopoulos, N.: Aerosol events in the broader Mediterranean basin based on 7-year (2000–2007) MODIS C005 data, Ann. Geophys., 27, 3509-3522, doi:10.5194/angeo-27-3509-2009, 2009.

Gkikas, A., Hatzianastassiou, N., Mihalopoulos, N., Katsoulis, V., Kazadzis, S., Pey, J., Querol, X., and Torres, O.: The regime of intense desert dust episodes in the Mediterranean based on contemporary satellite observations and ground measurements, Atmos. Chem. Phys., 13, 12135-12154, doi:10.5194/acp-13-12135-2013, 2013.

Papayannis, A., et al. (2008), Systematic lidar observations of Saharan dust over Europe in the frame of EARLINET (2000 – 2002), J. Geophys. Res., 113, D10204, doi:10.1029/2007JD009028.

Querol, X., Pey, J., Pandolfi, M., Alastuey, A., Cusack, M., Pérez, N., Moreno, T., Viana, N., Mihalopoulos, N., Kallos, G. And Kleanthous, S.: African dust contributions

to mean ambient PM10 mass-levels across the Mediterranean basin, Atmos. Environ., 43, 4266–4277, 2009.

The authors should review the literature in order to establish the extreme character of the event. More specific comments regarding the Introduction follow:

i) Page 1, Lines 11-14: "The visibility ... 10000 $\mu$g/m3." If I understand these are the results of the paper. If not provide the reference otherwise delete the sentence.

ii) Page 1, Lines 14-15: Some thing as previously if there is a reference, please provide it otherwise delete.

iii) Page 1, Lines 16-17: "Surprisingly ... models." Please provide information about the models. E.g. http://sds-was.aemet.es/?

iv) Page 1, Line 23 to Page 2, Line 7: How useful is to provide this breakdown of Section 3 in a short paper? The authors repeat themselves several times. I suggest deleting it and incorporating any additional information provided in this part (if any) to the relevant subsections.

5) Page 2, Lines 14-15: Although, there are not observations from the CUT-TEPAK AERONET sun-photometer during the event, there are observations (at least Level 1.5) from other sun-photometers in the region like AgiaMarina_Xyliatou and SEDE_BOKER. It is important to compare MODIS AOD retrievals with AERONET values in order to establish how good there are in the case of large AODs (>2). This is important, especially, from the moment that you are using MODIS AOD over Cyprus in Section 3.1.

Minor comments

6) Page 1, Lines 2-3: Which dust models, give details otherwise delete.

7) Page 2, Line 19: Add reference for MODIS AOD.

Levy, R. C., Mattoo, S., Munchak, L. A., Remer, L. A., Sayer, A. M., Patadia, F., and
Hsu, N. C.: The Collection 6 MODIS aerosol products over land and ocean, Atmos. Meas. Tech., 6, 2989-3034, doi:10.5194/amt-6-2989-2013, 2013.

8) Page 2, Lines 26-30 and Page 3, Lines 19-26: The authors present AOD and PM10 observations for several sites in Cyprus, but they do not discuss the differences between them, i.e. the spatially variability. Either discuss the spatial variability among the different sites or just present the data for Limassol. In the former case, I suggest to present the MODIS AOD maps (similar to Figure 1) instead of the AOD time series, probably zooming in Cyprus. Otherwise use the same time range and the same cities for Figures 2 and 4.

9) Page 3, Lines 9-11: Provide reference for the visibilities obtained from pilots.

10) Page 3, Line 23: "... and thus main contain errors." Ask the Air Quality Department of Cyprus for validated data. Otherwise, what's the point of presenting data in your manuscript which are of unknown quality?

11) Page 3, Line 30: Replace "Cyprus" with "Limassol", as the lidar observations are from Limassol and as you stated in Page 2, Lines 23-25 the Troodos Mountains were always visible during the dust storm, certainly not affected the same way with Limassol.

12) Page 3, Line 30: "A two-layer structure ... on 8 September," this is pure speculation. Either present observations or delete it.

13) Page 4, Lines 5-7: This is just a speculation, as you do not have any information about the vertical distribution of the dust. The authors should underline more this fact using a stronger word than assumption. The fact that the Troodos Mountains were not covered by dust means that possibly the dust layer was below 2 km, but it could be just 500 m (or lower) like the 9th September (Figure 5). On the other hand, Figure 6 shows through the backscatter coefficient that the layer was not homogeneous with the higher values at surface during the 9th September, so this could be the case for the 8th. At this point MODIS maps can give a hint about the area with the high AOD (even

saturated at 5), while also the AOD observations from MODIS/Terra could be useful to check the temporal variability. Certainly, this result although plausible can not be one of the main conclusions of the paper as it is based on assumptions which can not be verified. Instead the authors could use the MODIS AOD.

14) Page 4, Line 19: The word frequently is in contrast with just one case study mentioned in the next sentence. Either rephrase or use another reference for the altitude of Saharan dust plumes over Cyprus.

15) Page 4, Line 24: Why you do not use the same time interval, e.g. 18-20 UTC for all profiles shown in Figures 6, 7? Although, I do not expect large changes in the results (if any) the comparison between the four days will become straightforward.

16) Page 5, Lines 1-4 and Figure 6: For all four days the depolarization ratio reduces significantly below about 0.5 km. This means that surface aerosols are always a mixture of dust with pollution or is just an instrument artefact e.g. overlap function?

17) Section 3.6 and Figure 8: I do not think that this section and figure add something in the paper. I suggest to totally removing both. Otherwise, the authors should justify their utility.

Technical comments 18) Page 1, Line 24 and Page 2, Line 18: Replace 'imaginary' by 'imagery'.

---

## Author Comment (AC2) · 2 Sep 2016

**Dear Editor,**

We thank the reviewers for careful reading, and for taking the time to write the constructive reviews which helped us to re-think the concept of the paper, to re-evaluate the observation and to consider even new observations, and to rewrite the paper contents as a whole along all the suggestions by the reviewers. Because of all these changes, it makes no sense to present a revised version with marked text packages (indicating changes).

Before presenting step-by-step answers to the detailed comments, we want to summarize the main points of changes:

- **We changed the title and added new co-authors (modeling partners).**

- **We extended the abstract, more straight forward, more observational findings (numbers), almost no speculations.**

- **We changed the introduction, and now clearly state: what is new, what is the motivation for the paper, and that there will be a second complementary (modeling) paper dealing with this record-breaking dust storm. The first draft of the second paper was distributed in the beginning of August.**

- **This second paper is: Solomos et al., Extreme dust storm over Middle-East and the Eastern Mediterranean in September 2015: Modeling study with RAMS-ICLAMS, to be submitted to ACP.**

- **We provide a long section 2 on instruments and retrieval products, almost two pages instead of one paragraph (given in the submitted first version).**

- **We include new observation (visibility observation at three airports in Cyprus and quality-assured daily mean PM10 observations).**

- **We show radiosonde data and include a few modeling results (of the second paper) to show the failure of the models…. (Figure 4). We explain the failure (cloud and thunderstorm processes are not resolved, and also provide the most likely reasons for the enormous dust mobilizations, probably caused by a haboob and associated density currents and strong downbursts.**

- **We re-arranged the order of figures to make the full presentation and discussion more straight forward and exciting.**

- **We use Nicosia radiosonde temperature and humidity profiles now in the discussion of the 8 September observations including the dust layering.**

**Reviewer #1**

**Our answers in bold**

The paper describes an exceptional dust storm over Cyprus. Using lidar data, visibility studies, ground based PM10 measurements and several assumptions it is tried to characterize the event. This should help to improve transport models which – according to the authors – totally failed to predict this outbreak. The paper is clearly structured and easy to understand. It focusses on nothing but the dust storm – thus it is quite short. I did not find any errors.

**The paper is now much longer (14 pages of text now vs 6.5 pages of text of the submitted version), includes more observations and discussion about the consistency between the different observations.**

I think that the paper can in principle contribute to two aspects: validation of transport models or (optical) characterization of desert dust. In both cases the current version must be improved significantly.

**We agree and explain in the introduction how we cover that: One paper on observations, and one (follow-up) paper dealing with the simulations.**

According to the manuscript the main motivation is to provide information for the improvement of transport models. However, no strategy how to reach this goal is detailed, not even outlined. Which models are meant (in the introduction the authors mention 'state-of-the-art dust transport models' [plural], but only HYSPLIT is mentioned later)? Are the depolarization ratio and the lidar ratio useful parameters to improve a trans- port model? How can these parameters actually contribute to this goal? Would it be sufficient to determine only the backscatter coefficient and the PM10-concentration to check any improvement of the model? According to Section 3.6 wrong meteorological fields were primarily responsible for the bad forecast (not the numerical description of the microphysics of the model). Is it necessary to estimate the vertical profile of 8. September (this is really vague!) for this purpose or would an agreement for the other days be sufficient to check the improvement of the model? From this point of view I recommend to combine this paper with the 'follow-up'-paper mentioned on page 5 line 25. Then it would be possible to clearly demonstrate the role of the data for the improvement of the model, and to show whether it was successful or not.

**We clearly state in the introduction that we will have two papers, the first one (this one) will cover the observations, the second one the simulations and the research of the reasons for the major dust storm and why the forecast models failed. All observations are now discussed in a logic way with a minimum of speculations. However, some speculative assumptions cannot be avoided. They are carefully justified and discussed in section 3.1 and 3.2.**

The other aspect – I had expected this when I saw the paper the first time – is to study optical properties of desert dust. Though there are already many similar papers on dust of different deserts and different transport paths (mixing, aging), this paper could potentially be a useful contribution: it comprises the 'standard output' of advanced lidar systems (lidar ratio, depolarization ratio, wavelength-dependent extinction and backscatter coefficients),

provided that the accuracy of these parameters will be added. If the source region can clearly be identified this study can be a contribution towards a climatology of optical properties of different deserts.

**No, we do not like to follow the idea of the reviewer. We just want to present and document a record-breaking dust storm (based on observations). Such storms are rarely described in the literature (we state that), they may occur ones in 20 years in the Eastern Mediterranean. The second paper will deal with the context of the unusual meteorological conditions that triggered this dust storm. This basic concept we want to present.**

A combination of both aspects would certainly be the most attractive solution and together with the improvement of the transport model this paper could be a really interesting scientific study (different from many previous 'dust-papers') and would fit to ACP.

**As mentioned ….., with two papers (one dealing with the observations, one covering the simulation) we now have a good concept.**

By the way: it is surprising that more than 80% of the references are from the authors themselves. I am sure that there are more publications on these topics.

**We increased the list of references as a whole significantly and decreased the percentage of self-citations.**

A few minor comments:

• 1-12: 'hit' change to 'hits'

**changed**

• 2-14: What is CUT-TEPAK?

**Is now explained in section 2.1. Cyprus University of Technology… and then in Greek TEchologiko PAnepistimio Kyprou.**

• 3-10: 'observations from pilots':  where is this information actually coming from? What exactly do they provide?

**This information is removed, now we show measured visibilities.**

• 3-16: Is this from the Koschmider's formula?

**Yes, it is! We now provide much more explanations in an extra subsection 2.4 on visibility measurements.**

• 3-22: 'are not validated': What does this mean? What errors can be expected? Would it have been possible to validated these data (under which conditions)?

**Removed**

• 4-6: 'AOT close to 1': where is this estimate coming from?

**Changed, or better removed. It was estimated from the lidar observations.**

• 5-8: 'The technique applied ... is described': This should be briefly outlined here so that the reader immediately can understand which measurements and which assumptions are used.

**We extended significantly the description of the observational methods in section 2. We introduced subsections 2.1-2.4. We provide many references and provide uncertainty discussions.**

• 5-10: typo: 'extreme'

**Changed**

• Fig. 7: I would expect that the curves in Fig. 7 (dust concentration) are proportional to the backscatter profiles shown in Fig. 6. At least for 10. September this seems not to be the case. This emphasizes the need to better explain the way how the mass concentration is determined (see previous comment).

**All profiles are ok. The old figure 7 (now figure 9) shows the pure dust mass profiles, whereas the profiles in old Figure 6 (now figure 8) showed the total aerosol backscatter profiles (dust plus non-dust components)**

**Reviewer #2**

Extreme dust storm over the eastern Mediterranean in September 2015: Lidar vertical profiling of desert dust at Limassol, Cyprus.

1 General comments

This article describes an exceptional dust event observed in Cyprus. The authors combined remote sensing from ground and space with ground-based in-situ aerosol measurements and models to give a comprehensive overview of the dust plume. The paper is rather descriptive, but the described methods are sound and the data set is unique. Therefore, I would recommend the paper to be published in ACP. However, there are some fundamental points that need to be addressed before publication. My specific comments and technical corrections are given below.

**The paper deals with a record-breaking dust storm that may take place ones in 20 years. We bring together many state-of-the-art observations including excellent Raman/polarization lidar observations. So all this unique, and should be enough to justify publication. Of course we got the message, and improved the introduction to provide a proper motivation for this paper (discussion of observations) and that there will be a second paper dealing with modeling of this event.**

2 Specific comments

Both, introduction and conclusions, should be reworked. The introduction is giving results described later in the text, which is not appropriate. Besides, in a rather short and straightforward paper like this, a detailed description of the structure of the paper is not necessary. I would suggest to present a stronger, more concise motivation. Also, the state of knowledge on Middle East dust is not discussed and should be included.

**Yes, we agree, and improved the introduction and conclusion sections accordingly. However, we leave out to discuss the state of knowledge of Middle East dust.**

The conclusions are rather a summary of the results. One of the review criteria for publication in ACP is the following: "Are substantial conclusions reached?" Please consider this point, which is the main weakness of this manuscript in my opinion.

**We got the message and improved the introduction and conclusions.**

What I'm also missing are information on the difference between local time and UTC. Besides, it would be very beneficial to include more details on the model forecast. The failure of the model is mentioned a few times, but what was actually forecasted?

**Is now given in Figure 1, caption (EEST: Eastern European Summer Time, three hours in front of UTC), and also repeated in the text.**

Some more detailed comments:

page 1, line 3 Please include a better discussion about the models failing to predict the event in the main text. You highlight it in the abstract and it is repeated in the main text, but it is not shown.

**Is now given in the Introduction.**

introduction The first paragraph of the introduction is a summary of results. Please remove. Also I'm missing some lines motivating this study. It is remarkable, but why is it important to study such cases in detail? Please also discuss literature on Middle East dust.

**We improve the Introduction as a whole, provide motivating points for reporting this record-breaking dust storm, stating our two-paper concept, providing literature hints on aerosol climatologies in the Mediterranean, etc….**

page 2, line 2 replace "accurately determined" by "estimated"; I don't think an estimation "around 500 m, but clearly below 750 m" should be called accurate.

**We changed the text accordingly. We estimate the visibility now to be 500-600m.**

page 2, line 4 Also here I have some concerns about the "good accuracy". You don't give uncertainties in your estimate, but a range of values. Which is ok, but a range of 3000 µg/m3 should not be called accurate. You present a simple and very nice method to estimate the dust load, but the emphasis here is on "estimate". It is not a precise measurement and should not be presented as such.

**We agree and changed wording.**

page 2, line 10 Why were data at 1064 nm not used for the dust characterisation, for example to calculate the Ångström exponent (only for time-height plots, figure 5)?

**We tried, but the result was not reasonable, we speculate that the photon-detecting APD prohibits a good retrieval 1064nm backscatter signals.**

page 2, line 12 Which of the cited publications describe the set-up used for this work?

**We extended the instrument descriptions in Section 2. Provide now detailed information in four subsections and detailed references for the methods and product uncertainties.**

page 2, line 13 Which of the cited publications describe the data analysis used in this study? It would be helpful to at least give the main points of the analysis and specify which reference describes what.

**As just mentioned, we do it now in a better way.**

page 3, lines 17-18 Could you please discuss the influence of humidity and consequent hygroscopic growth of the dust particles on the dust mass concentration?

**Yes we do that now in the result section 3.2. But the relative humidity was at all below 50% so there is no water uptake effect, and then we have dust which is quite hydrophobic**.

page 4, line 12 I would interpret "morning" and "noon" as indicators of local time. But either way, 14:20 is afternoon. It's a small detail, but can be confusing if you are not familiar with the time zone of Cyprus. Rather be specific.

Local noon is 12 local time.

**We provide now local time information at several places (EEST).**

figure 6, caption How did you obtain the uncertainties?

**We give this information now in Section 2.1. There, we provide references to the retrieval methods and the uncertainty estimations. We obtain the errors by assuming the error propagating law.**

page 5, line 10 A "front" is a distinct line (or surface); the front of the plume. The passage of it should not take three days. I suggest you rephrase by replacing "dust front" by "dust plume" or similar.

**We are now careful with wording.**

page 5, line 13 Who defined this threshold? And why?

**The statement is removed.**

page 5, line 20 Predicted by which model(s)? What did it/they predict for 35∘? And for 33∘? conclusion It is a summary rather than conclusions. Try to make a stronger case for your findings. What is the contribution of this work to existing knowledge? And how does it link back to your motivation?

**We improved the Introduction and the conclusions, give better motivation, give real conclusions. Not just summary of the paper, and say that we will have a second paper, focusing on modeling of this event.**

3       Technical corrections

**All of the following remarks were considered before we started to rewrite the entire paper, section by section**

page 1, line 20 replace "re-analyze" by "re-analyzing"

page 1, line 24 replace "imaginary" by "imagery"

instrumentation I suggest to rename this section to "Instrumentation and methodology".

page 2, line 18 replace "imaginary" by "imagery"

page 2, line 20 replace "dust plumes were partly" by "parts of the dust plumes were"

figure 2 Why are there two dots at the same location each day? Are those from MODIS on Aqua and on Terra, respectively? Please specify this in the text or caption.

**Yes, from MODIS on AQUA and TERRA. We state that now in the respective figure (new figure 6).**

figure 5, caption In my opinion the following part of the caption should be included in the main text body as part of the discussion of the figure: "The signals backscattered by dust in the elevated layers above 1000-1500 m height are partly strongly attenuated by the desert particles occurring below 1500 m. As a consequence, the elevated layers are mostly given in blue and green instead of red (as it would be the case after the correction of the attenuation effect)."

**Done**

page 3, line 31 Please include lines 3 and 4 from page 4 (starting with "Unfortunately, ..." ending with "... detection units.") after the sentence ending with "... the highest dust load.".

page 3, line 31 I don't think it is obvious, as you don't show lidar profiles from that day. You could rephrase this sentence, e.g. "Judging from the MODIS observations in Fig. 1., the rather thick dust layer reached 1000-1500 m height on 8 September."

**We now include radiosonde information of 8 September. These data perfectly corroborate our 'speculation'. So, there was definitely a 1.5 km thick and well-mixed layer (with base at surface on 8 Sep, noon) causing this optical depth of 6-9. In this**

**estimation of AOT we take the estimate of the surface extinction coefficient to 6000 Mm-1 and assume a 1.5 km thick well-mixed layer. That's it. And we end up with an AOD of 9. Lidar and radiosonde data are in perfect harmony for the entire period from 7 to 11 September. This motivated us to use the radiosonde profiles alone to estimate the layering on 8 September. We find, this is fully justified. So, we do now use the 8 September temperature and humidity profiles to explain the dust layering and that the dust was well mixed from the ground up to 1500 m height (as inidcated by the humidity profiles on 8 Sep) and caused and optical depth of 6-9. All this is now discussed in Section 3.2.**

page 4, line 12 replace "session" by "sessions"

page 4, line 19 replace "is" by "was"

figure 6 Please change the labelling on the x-axis of the depolarisation ratio plot.
It is hard to tell the numbers apart. You could leave major ticks at 0.2 and 0.4 (with labels) and use minor ticks at 0.1 and 0.3 (without labels).

**Done**

figure 6, caption The following parts should be included in the main text body rather than the caption: "The Raman lidar method is applied.", and " Retrieval uncertainties are of the order of 10% (backscatter coefficient, depolariza- tion ratio), 25% (extinction coefficient), and 30% (lidar ratio)."

**Since we avoid to show error bars to make the figures not too busy, we left the information about uncertainties in the captions (but also explain that in the text).**

figure 6, caption Add reference to "Raman lidar method". [now in text body]

page 5, line 10 replace "extrem" by "extreme"
page 5, line 15 Please specify length of trajectories in days in the text. It is in my opinion not enough to include it in the caption of figure 8.
page 5, line 19 replace "dust advection" by "air mass transport"
figure 7, caption Please move the following part of the caption to the main text body: "A dust particle mass density of 2.6 g/cm3 is assumed in the retrieval. The overall uncertainty is 30% and mainly caused by the uncertainty in the dust volume-to-extinction ratio (extinction-to-volume conversion factor)
assumed to be 0.8 $*$ 10−6  m."
figure 7, caption Add references for the mass density, conversion factor and uncertainty estimation. [now in text body]

**Reviewer #3:**

Review of the paper "Extreme dust storm over the eastern Mediterranean in September 2015:  Lidar vertical profiling of desert dust at Limassol, Cyprus" by Mamouri et al. submitted to Atmospheric Chemistry and Physics.

This short paper describes the case of an extreme dust plume occurred over Eastern Mediterranean lasting for few days and observed with ground based measurements at Cyprus. In addition satellite products from MODIS are used. The paper is within the scopes of Atmospheric Chemistry and Physics as it presents a rare (may be unique) dust event. However, in its current version needs major revision before it can be accepted for publication. The two main reasons is the lack of uncertainties throughout the paper and the presentation of speculative results (although not necessarily wrong) in the aim to characterize the event especially for the 8th September when lidar data were not available. For more details see the comments below

Major comments

1) Cyprus has four WMO stations (https://www.wmo.int/cpdb/dashboard/index/countryCode:CYP). Why you do not present visibility results from them?   At least 3 (Akrotiri, Larnaka, and Paphos) of them take visibility observations, with Akrotiri being next to Limassol (~25 km). The data should be available at least through the national meteorological service of Cyprus.  Although, the photographs in Figure 3 are indicative of the low visibility occurred during the noon of 8th and the contrast with 9th September, still they do not follow exactly the WMO guidelines especially when visibility is poor, as established in WMO Guide to Meteorological Instruments and Methods of Observation (http://library.wmo.int/pmb_ged/wmo_8_en-2012.pdf). The visibility results will permit also an assessment of the variability of the dust extinction-to-mass conversion factor given the observations of PM10 for the 3 sites providing both types of observations.

**Thank you for hint! We now show the visibility observations from the three airports mentioned above (new Figure 7). However, to analyze photos with a lot of buildings is to our opinion much more accurate than to estimate visibility at very unusual dusty conditions! And later, when we compare the visibility-based mass concentrations with PM10 values, we clearly get the impression that the airport visibility estimates are quite erroneous (by a factor of 2 too low visibilities) for 8 September. We discuss this in section 3.2).**

2) For a short paper like this presenting an extreme event, what I was hoping to find was uncertainties to all the observations and of course the factors used from previous studies. Just providing mean values is not enough in order to establish the importance and the uniqueness of this event, especially for the values mentioned in the abstract and conclusions. Although, the authors provide estimations in the legends of Figures 6 and 7, it would be better to introduce them in the manuscript and the figures. While there are not uncertainties presented regarding the results of 8th September.

**We expanded the instrumentation section 2, now with four sub-sections, and provide many references to the techniques and the uncertainties, and provide a good overview of all the uncertainties. We do not show error bars in the figures (now 8 and 9) to avoid overloading. We prefer to provide uncertainty information in the captions.**

3) In the abstract and the conclusions you present mass concentrations values esti- mated from visibility by using a typical dust extinction-to-mass conversion factor. Also, the AOD values of 6-10 are based on assumptions about the vertical distribution. Although, in both cases the values are logical (according to the analysis presented in Section 3), simply they are not measured, so I strongly suggest to present in the abstract and the conclusions only the observations by adding the phrase "with possibly higher values occurred the 8th

September" or something similar. Especially, in the case of mass concentration there are measurements and PM10 (or PM2.5) is what you find in the literature. For this reason is more appropriate to present the PM10 measurements in the abstract/conclusions.

**We changed the abstract accordingly, and removed speculative values.**

4) The Introduction needs rewriting, especially paragraphs 1 and 3. What's the point to present the results of the study already in the Introduction section in a short paper like this? On the other hand, there is no reference at all about climatological and extreme event studies for the region. At least, there are studies covering the Eastern Mediterranean dealing with AOD, lidar measurements and PM10. Example of relevant papers (certainly a non exhaustive list) can be found below:

Basart, S., Pérez, C., Cuevas, E., Baldasano, J. M., and Gobbi, G. P.: Aerosol characterization in Northern Africa, Northeastern Atlantic, Mediterranean Basin and Middle East from direct-sun AERONET observations, Atmos. Chem. Phys., 9, 8265-8282, doi:10.5194/acp-9-8265-2009, 2009.

Gkikas, A., Hatzianastassiou, N., and Mihalopoulos, N.: Aerosol events in the broader Mediterranean basin based on 7-year (2000–2007) MODIS C005 data, Ann. Geophys., 27, 3509-3522, doi:10.5194/angeo-27-3509-2009, 2009.

Gkikas, A., Hatzianastassiou, N., Mihalopoulos, N., Katsoulis, V., Kazadzis, S., Pey, J., Querol, X., and Torres, O.: The regime of intense desert dust episodes in the Mediterranean based on contemporary satellite observations and ground measurements, At- mos. Chem. Phys., 13, 12135-12154, doi:10.5194/acp-13-12135-2013, 2013.

Papayannis, A., et al. (2008), Systematic lidar observations of Saharan dust over Europe in the frame of EARLINET (2000 – 2002), J. Geophys. Res., 113, D10204, doi:10.1029/2007JD009028.

Querol, X., Pey, J., Pandolfi, M., Alastuey, A., Cusack, M., Pérez, N., Moreno, T., Viana, N., Mihalopoulos, N., Kallos, G. And Kleanthous, S.: African dust contributions to mean ambient PM10 mass-levels across the Mediterranean basin, Atmos. Environ., 43, 4266–4277, 2009.

The authors should review the literature in order to establish the extreme character of the event.

**The introduction is completely re-written. As mentioned above, the introduction now contains the motivation of the paper, the hint that there will be second modeling paper, and we provide a few references on aerosol conditions over the Eastern Mediterranean, and give definitions of strong and extreme dust outbreaks. And an outbreak causing an optical depth of 0.8 is already an extreme dust outbreak, and we had 6-9!...which may happen once in 20 years…**

More specific comments regarding the Introduction follow:

i) Page 1, Lines 11-14: "The visibility ... 10000 µg/m3." If I understand these are the results of the paper. If not provide the reference otherwise delete the sentence.

**All this re-written and presented in a clear way. The link between visibility and extinction is well known,,, and between extinction and volume or mass concentration is also not new and describe in section 2.1. now explicitly.**

ii) Page 1, Lines 14-15: Some thing as previously if there is a reference, please provide it otherwise delete.

**We changed it.**

iii) Page 1, Lines 16-17: "Surprisingly ... models." Please provide information about the models. E.g. http://sds-was.aemet.es/?

**We change it in the introduction accordingly and provide the link to the sds-was web page.**

iv) Page 1, Line 23 to Page 2, Line 7: How useful is to provide this breakdown of Section 3 in a short paper? The authors repeat themselves several times. I suggest deleting it and incorporating any additional information provided in this part (if any) to the relevant subsections.

**As mentioned we re-wrote the entire paper….**

5) Page 2, Lines 14-15: Although, there are not observations from the CUT- TEPAK AERONET sun-photometer during the event, there are observations (at least Level 1.5) from other sun-photometers in the region like AgiaMarina_Xyliatou and SEDE_BOKER. It is important to compare MODIS AOD retrievals with AERONET values in order to establish how good there are in the case of large AODs (>2). This is important, especially, from the moment that you are using MODIS AOD over Cyprus in Section 3.1.

**We try to compare the MODIS observations with our own lidar observations (section 3.2). We do not include other stations. The dust outbreak was so inhomogeneous, what does it help to include other stations? The uncertainty of MODIS values is clear, about 0.15 times AOT.**

Minor comments

6) Page 1, Lines 2-3: Which dust models, give details otherwise delete.

**All of them! This is now better stated in the introduction, pointing to the sds-was web page.**

7) Page 2, Line 19: Add reference for MODIS AOD.

Levy, R. C., Mattoo, S., Munchak, L. A., Remer, L. A., Sayer, A. M., Patadia, F., and

Hsu, N. C.: The Collection 6 MODIS aerosol products over land and ocean, Atmos. Meas. Tech., 6, 2989-3034, doi:10.5194/amt-6-2989-2013, 2013.

**Thank you, we now include Levy, 2013 and also one of 2010. We provide information on the uncertainty in section 2.2, is about 0.15 AOT.**

8) Page 2, Lines 26-30 and Page 3, Lines 19-26: The authors present AOD and PM10 observations for several sites in Cyprus, but they do not discuss the differences between them, i.e. the spatially variability. Either discuss the spatial variability among the different sites or just present the data for Limassol. In the former case, I suggest to present the MODIS AOD maps (similar to Figure 1) instead of the AOD time series, probably zooming in Cyprus. Otherwise use the same time range and the same cities for Figures 2 and 4.

**We kept these good suggestions in mind and changed the presentation now. We combined the old Figures 2 and 4 to the new Figure 6. But we prefer numbers and time series here, even if they were clearly biased on 8 September (when AOT was higher than 5).**

9) Page 3, Lines 9-11: Provide reference for the visibilities obtained from pilots.

**We removed this. We just found it in the internet. Now we have visibility observations, that is better solution.**

10) Page 3, Line 23: "... and thus main contain errors." Ask the Air Quality Department of Cyprus for validated data. Otherwise, what's the point of presenting data in your manuscript which are of unknown quality?

**Yes, we did, but asked us (how good are these data during such an extreme dust event…). Every observation is somehow optimized to work well at usual and up to some extreme conditions. But here at these rather unusual conditions? We do not really believe in any (validated or not validated) in situ observation. Furthermore, the only validated data that DLI provides are daily mean values, following the EU guidelines. We used the hourly means at 4 DLI stations to highlight the spatial-temporal variability and the density of the event. Nevertheless we discuss the deviations between the validated daily means and the daily means calculated from the non-validated hourly means and found deviations of 50%. That is discussed now in section 2.3.**

11) Page 3, Line 30: Replace "Cyprus" with "Limassol", as the lidar observations are from Limassol and as you stated in Page 2, Lines 23-25 the Troodos Mountains were always visible during the dust storm, certainly not affected the same way with Limassol.

**We made the discussion more save… (sections 3.1 and 3.2)**

12) Page 3, Line 30: "A two-layer structure ... on 8 September," this is pure speculation. Either present observations or delete it.

**As already given above, we now include radiosonde information of 8 September. These data perfectly corroborate our 'speculation'. So, there was definitely a 1.5 km thick and well-mixed layer (with base at surface on 8 Sep, noon). And by using the visibility-related extinction coefficient of 6000 Mm-1 we end up with an optical depth of 9. All in all, lidar and radiosonde data are in perfect harmony for the entire period from 7 to 11 September. This motivated us to use the radiosonde profiles alone to estimate the layering on 8 September. We find, this is fully justified. So, we do now**

**use the 8 September temperature and humidity profiles to explain the dust layering and that the dust was well mixed from the ground up to 1500 m height (as inidcated by the humidity profiles on 8 Sep) and caused and optical depth of 6-9. All this is now discussed in Section 3.2.**

13) Page 4, Lines 5-7: This is just a speculation, as you do not have any information about the vertical distribution of the dust. The authors should underline more this fact using a stronger word than assumption. The fact that the Troodos Mountains were not covered by dust means that possibly the dust layer was below 2 km, but it could be just 500 m (or lower) like the 9th September (Figure 5). On the other hand, Figure 6 shows through the backscatter coefficient that the layer was not homogeneous with the higher values at surface during the 9th September, so this could be the case for the 8th. At this point MODIS maps can give a hint about the area with the high AOD (even saturated at 5), while also the AOD observations from MODIS/Terra could be useful to check the temporal variability. Certainly, this result although plausible can not be one of the main conclusions of the paper as it is based on assumptions which can not be verified. Instead the authors could use the MODIS AOD.

**As mentioned, we now include radiosonde information of 8 September….. These data indicate a well-mixed 1.5 km thick lower dust layer. And with surface extinction coefficients of 4000-6000 Mm-1 we end up with AOT of 6-9. And this also in agreement with MODIS. MODIS stops when the AOT is largere than 5. And there are many MODIS pixels (areas) with higher AOT than 5, as discussed in section 3.2. MODIS does not provide data higher than 5. We use MODIS TERRA and AQUA data and do not see strong temporal variability…**

14) Page 4, Line 19: The word frequently is in contrast with just one case study mentioned in the next sentence. Either rephrase or use another reference for the altitude of Saharan dust plumes over Cyprus.

**We changed it**

15) Page 4, Line 24: Why you do not use the same time interval, e.g. 18-20 UTC for all profiles shown in Figures 6, 7? Although, I do not expect large changes in the results (if any) the comparison between the four days will become straightforward.

**The lidar is not that powerful, so we need long signal averaging times. And if we have the chance to use longer time periods (with comparably homogeneous aerosol conditions) we should take the opportunity.  So, we leave Figure 8 and 9 (old figures 6 and 7) as they were…. regarding signal averaging time intervals.**

16) Page 5, Lines 1-4 and Figure 6: For all four days the depolarization ratio reduces significantly below about 0.5 km. This means that surface aerosols are always a mix- ture of dust with pollution or is just an instrument artefact e.g. overlap function?

**The depolarization profiles are fine, and show the pollution impact . We discuss that at the end of Section 3.2 (based on numbers for mass concentrations).**

17) Section 3.6 and Figure 8: I do not think that this section and figure add something in the paper. I suggest to totally removing both. Otherwise, the authors should justify their utility.

**We removed all this (a bit), but want to show at least one HYSPLIT result so that the reader get at least a rough idea about the air mass transport from the Middle East on these days…**

Technical comments 18) Page 1, Line 24 and Page 2, Line 18: Replace 'imaginary' by 'imagery'.

**Done**

**Reviewer #4:**

This brief study presents lidar measurements acquired during an extreme dust storm which swept across the Middle East in September 2015. The lidar observations are high quality and are fascinating. I have no particular issues with the data process- ing etc... However, I think that the scientific content of the paper is weak. I do not recommend publication in ACP, for the reasons detailed below.

I believe the paper would be a better match for AMT.

**This is a surprising statement! … and we do not agree.**

**To our understanding, ACP is the journal for presenting scientific results which includes atmospheric observations. AMT is the journal for presenting new techniques and new instruments. And we present observations.**

General Comments

1. In particular, I am missing an analysis of the synoptic situation leading to the dust storm, and how this situation evolved and led to the demise of the dust storm as observed with MODIS and the lidar in Limasol. I am also missing a detailed analysis of the processes leading to the dust emission and transport, as well as the activated source regions. In several instances, the authors state that dust transport models failed to predict this record dust event, without giving precision on the models, their resolution, etc. . . An analysis of such processes would be extremely useful to understand why the transport models did fail, if they did.

**As now outlined in detail in the abstract (already) and in the introduction, we will have a series of two paper: part one (this paper) on observations and part two (all the modeling aspects). In the second paper, the meteorological conditions will be discussed and the specific reasons for this huge dust storm will be outlined. However, we give a short meteorological explanation already in the introduction of this paper and then in section 3.2.**

2. I also find the interpretation of the HYSPLIT back trajectories to lack insightful analysis. The authors find them to be in contradiction with their dust observations, and ascribe the differences to erroneous meteorological fields provided by global scale models. This may well be the case but the main issue is that HYSPLIT cannot not deal with turbulence and transport in turbulent planetary boundary layers (PBLs). However, this is not verified in the

present study and one wonders what is the worth of including these back trajectories in the analysis (?).

**The HYSPLIT trajectories are not very useful, we agree. Nevertheless, we show one HYSPLIT plot to provide a rough idea about the main features of dust transport and some hints on dust sources.**

3. Emphasis is put in the abstract on the supposedly large AOD and mass concentra- tion during the event. However, these values are not observed on 8 September (the day when the storm was most intense), and are only speculative, extrapolated from indirect measurements, or based on assumptions. They can be discussed in the text, but should not be emphasized that much in the Abstract and in the conclusion (where they are presented as results, see p5, l 28: "Dust AOT values of the order of 6-10 oc- curred over Cyprus [. . .]"). Furthermore, the very large AOD values (6-10) inferred on that day from pseudo-lidar measurements and yet MODIS retrievals on the same day are thought to be biased (p2, l30).

**We removed speculative values from the abstract and the conclusions (completely re-written).**

**As mentioned already above, we now include radiosonde information of 8 September. These data perfectly corroborate our 'speculation'. So, there was definitely a 1.5 km thick well-mixed layer from the ground to 1.5 km height. And when we use the visibility of 500m and therefore extinction coefficients of 6000 Mm-1 then we simply have to conclude that the AOT was 9 (for 4000 Mm-1 we get an AOT of 6). Lidar and radiosonde data are in perfect harmony for the entire period from 7 to 11 September. So, we do not see any reason not to use the 8 September radiosonde temperature and humidity profiles (alone) to explain the dust layering on 8 September. All this is now discussed in a reasonable and consistent way in section 3.2.**

4. The 2-layer structure observed by lidar on 7 September suggests different emission source regions and transport regimes from Middle East, as is the case in other parts of the world, like the Sahara. This structure likely relates to differential advection and PBL dynamics over the emission regions. Knowledge of these processes would be extremely useful for a comprehensive analysis of the back trajectories.

**Yes, the two-layer structure indicates different dust sources and different transport regimes. More insight into the basic meteorological processes are given in the follow-up modeling paper (part 2, simulations).**

Minor comments

P1, l 14: where does the number 1000 $\mu$g m-3 come from?

**The Introduction is completely rewritten, the statement is removed.**

P1, l 17: which models do you refer to?

**All models failed. We now provide the internet link to the sds-was.aemet.es web page, where you can see that.**

P1, l 24: imagery

**Improved.**

P2, l20: […] we in some parts so dense […]

**Changed**

P2, l 24-25: yet you show later on that dust plumes were observed by lidar above that height. . . Does this mean that the dust plume transported at higher altitude was less optically thick?

**Yes the second layer shows AOTs of 0.5 on 7 and 9-10 September.**

P2, l 30: why are they biased?

**We averaged numbers <5.0 (good and validated observations) and many values of 5.0 (not observed, but just set to 5.0, although the true values were much higher…). Then we averaged all these numbers….**

P3, l 24-26: how do you know particles bigger that 10 $\mu$m are transported to Cyprus? Furthermore, what is the influence of marine aerosols on these measurements? Would not they influence the surface mass concentration measurements (Limasol is close to the sea)?

**We changed the text. Provide a discussion. Kandler et al. (2009, 2011) clearly shows that total dust mass concentration is always larger (by a factor of 1.2-1.8 after long-range transport) than the PM10 mass concentration. We corroborate that with our study (and mention that in section 3.2) So, there are larger particles. We also discuss the negligible impact of marine and urban particles in the case of a huge dust storm.**

P3, section 3.3: in this section, nothing is said about the high backscatter values observed in the lower layer (island or marine PBL?). Do not you expect dust to also be present in the lower layers as the result of entrainment at the top of the PBL?

**We discuss that now, too (at the end of section 3.2). Yes, there is always pollution aerosol besides dust aerosol at low heights….and yes, there is dust at surface level…**

P3, l30: regarding the 2 layer structure: you do not have lidar data, but do you have met soundings to show the suggested layering?

**Thank you for the hint. We plotted all available Nicosia radiosonde profiles of temperature and humidity, we found nice agreement between aerosol layering and temperature and humidity gradient changes…, text book like. And these features seen for all lidar measurement days (7, 9-11 September) corroborate our conclusions on the layering on 8 September. The two-layer structure was there all the time. The lowest layer was well mixed on 8 Sep. from the surface up to 1.5 km height. According to the visibility study on 8 September the dust mass concentration was extremely high and caused AOTs of 6-9.  All this is given in the discussion in section 3.1 and 3.2.**

[revised manuscript text omitted]

---

## Referee Report (RR1)

**Review of manuscript number: acp-2016-354**

**Extreme dust storm over the eastern Mediterranean in September 2015: Satellite, lidar, and surface observations in the Cyprus region**

September 14, 2016

**1 General comments**

This article describes an exceptional dust event observed in Cyprus. The authors combined remote sensing from ground and space with ground-based in-situ aerosol measurements to give a comprehensive overview of the dust plume. I would recommend the paper to be published in ACP due to the extreme conditions described. However, there are some points that need to be addressed before publication. My specific comments and technical corrections are given below.

**2 Specific comments**

I found some parts difficult to follow. There were results mentioned in the methodology part and vice versa. It would be beneficial if you could work on the structure of the manuscript.

Some more specific comments:

**page 2, line 35** Consider including *Preißler et al.* (2011).

**page 4, line 5** What output does MODIS provide for AOT>5? Are those data points set to 5? Please include this information here (it is mentioned in the results section).

**page 4, line 5** move "On 8 September, this value was frequently exceeded." to results section

**page 4, line 12-13** move "We found ... September)." to results section

**page 5, line 1-3** move "During the strong dust ... can be neglected." to results section

**page 5, line 25** Could you please explain why you chose a point 160 km East of Limassol over the sea as trajectory end-point? Is it because the trajectories ending over the island failed to capture the observed dust transport?

**page 5, line 25** Do you mean between 8 and 9 UTC on 8 September? How do you know without lidar observations and radiosounding profiles at 6 and 12 UTC?

**3 Technical corrections**

**abstract** Please provide full name for MODIS and EARLINET.

**page 1, line 10** "close to 8000 $\mu$g/m$^3$" or "near 8000 $\mu$g/m$^3$"

**page 1, line 21** remove ">"

**page 3, line 8** change "belongs to" to "is part of"

**page 3, line 16** remove last "and"

**page 5, line 20** change "a coarse idea information" to "a rough idea" or "an impression"

**page 5, line 22** figure and caption give arrival height of 2.5 km, text gives 1.5 km for lower dust layer

**page 5, line 23** add space after "(RH)"

**page 5, line 20-25** I find it hard to follow this part due to a number of parentheses (and at least one too many if I'm not mistaken). Please rephrase, incorporating the information in parentheses in the text.

**figure 5, caption** change "at the roof" to "from the roof"

**figure 6, caption** move everything from "The AOTs are..." to main text

**figure 6, map in panel c** increase font size, it is very hard to read

**page 8, line 15** "Kandler et al., 2011" in parentheses

**page 9, line 4** use either "wrong visibility estimations" or "a wrong visibility estimation"

**page 9, line 4** change "unusual very" to "very unusual" or just "unusual"

**page 9, line 7-8** there is a verb missing after "... so that the..."

**page 9, line 7** change "volumne" to "volume"

**page 9, line 23** change "visbility" to "visibility"

**page 9, line 31** change "oberved" to "observed"

**page 10, line 17** change "A more ... profiles" to "More ... profiles"

**page 11, line 8** change "keeping" to "taking"

**page 11, line 16** change "a variety optical" to "a variety of optical"

**page 11, line 20** replace "strength" by "intensity"

**page 11, line 27** change "improved" to "improve"

**page 11, line 34** remove "profile"

**References**

Preißler, J., F. Wagner, S. N. Pereira, and J. L. Guerrero-Rascado (2011), Multi-instrumental observation of an exceptionally strong Saharan dust outbreak over Portugal, *Journal of Geophysical Research: Atmospheres*, **116**, D24,204, doi: 10.1029/2011JD016527.

---

## Referee Report (RR2)

**Review of**

**Extreme dust storm over the eastern Mediterranean in September 2015: Satellite, lidar, and surface observations in the Cyprus region by Mamouri et al. [revised version]**

The paper describes an exceptional dust storm over Cyprus. Using lidar data, visibility studies, satellite data, ground based $PM_{10}$ measurements and several assumptions it is tried to characterize the event. This should help to improve numerical transport models which failed to predict this dust outbreak.

The authors completely re-wrote the first version of the manuscript taking into account the comments of the review(s), in particular they spent some efforts in minimizing the 'speculative part' as far as possible. This is very much appreciated especially as this is in principle not a simple task because of the large uncertainties of the involved parameters/quantities/measurements. So there is an inherent difficulty to detect any inconsistency. I don't think that further improvements can be achieved in view of the available (and missing) data sets. The results are useful and necessary to support the second paper that is under preparation according to the authors. It will be interesting to see to what extent the disagreement of the lidar derived and the modelled profiles (Figure 4) can be reduced.

Minor comments and notes

- 3/29: 'The uncertainties in all the optical properties...': Here, one of the papers by Gasteiger and colleagues should be cited who performed a lot of quite detailed investigations on this topic (desert dust, volcanic ash).

- 4/12: '$\pm$ 50': Unit is missing.

- 5/11: Fig. 1: Please mark Limassol in at least one of the panels

- 5/29: Why are the lidar measurements restricted to day-time measurements: lack of personnel? I assume the system does not run unattended?

- 6/8: 'and 600 $\mu g/m^3$...'. When averaging over 3 hours this is certainly a lower limit. From Fig. 3a it can be seen that during the first hour the concentration was significantly larger. So it was indeed a strong event!

- 7/17: 'Therefore the area mean values...' What does this mean? Is it – due to the fact that the maximum retrievable values might be exceeded – sort of an estimate of the lower limit?

- 8/16: Paragraph starting with 'To check...': I doubt that it is possible to determine a relationship between $PM_{10}$ and TSP taking into account the uncertainties of the contributing variables/measurements (during this episode), and I don't understand the message of the related discussion. The authors found an 'excellent agreement' with Kandler's values. On the other hand they present arguments against the assumed values (stating that either $c_{v,d}$ or $r_{vis}$ is wrong). If however $r_{vis}$ is wrong this would feedback to the estimated extinction coefficient and the mass concentration. Some additional explanations would be helpful to avoid possible confusion.

- 10/7: 'termine': typo.

---

## Author Response (AR2)

**Dear Editor, Dear Reviewers!**

Thank you again for taking the time for careful reading. We included the majority of your suggestions in the revised manuscript. They improved the paper again…

The main points of improvement are:

- We analyzed all MODIS overflights (Limassol overpasses) from 2001 to 2015 (about 5000 overpasses) and carefully created a terra-MODIS (2001-2002)-aqua-MODIS (2003-2015) AOT time series for Limassol (considering one observation per day, all in all 2018 daily observations). Then we checked this MODIS AOT data set for strong and extreme dust events. And as we already speculated, the extreme high 8-September 2015 AOT was by far the highest within this 15 year period. This is now shown in the introduction. We present a new figure (Fig.1) based on this 15-year AOT data set. We think that an extraordinary event (this September 2015 dust event) needs an extraordinary introduction with this figure as a highlight.

- We also performed a careful MODIS-AERONET AOT comparison by using AERONET data from Agia Marina/Nicosia (for moderate AOTS of 0.5-1.0) and the Weizmann Institute (Israel, AOT>2.5) for the September 2015 dust period. In addition, we used Limassol AERONET data (AOT>3.0, April 2013). We found a clear systematic overestimation by MODIS of the order of 0.5-1.5 for high AOTs (>2.5). This is discussed in the methodology section 2.2.

- We tried again to be less speculative regarding the peak AOT on 8 September, we simplified the discussion (following the suggestion of the reviewer who asked for major revisions) by assuming a well-mixed dust layer up to 800 m or 1500 m. With an extinction coefficient of 6 km-1 we end up with AOTs of 4.8 to 9. These value corroborate the MODIS data, which suggest AOT>5.0. We feel that such a speculative discussion is tolerable (not to say necessary) in a case of such an extraordinarily strong, unique dust storm.

- We integrated new literature mentioned by two reviewers.

**Step-by step answers (our answers in bold)**

**Editor's comment:**

I also have a comment to add: You refer to the occurrence rate of such heavy dust events at various occasions in the text. Where does this estimate come from? I think you should either back up this claim with references/measurements or omit it as it is rather speculative. Are the time series of ground-based PM10 observations at Cyprus (or in the region) long enough to assess the occurrence rate of such extreme dust events?

**As mentioned above, we analyzed the MODIS data from 2001-2015, and calculated area mean AOTs around Limassol (all data within 25 km radius around city center), we then calculated the mean value and the threshold value for a strong dust case (defined by an AOT>0.70 up to 1.15, mean plus 2 times standard deviation) and an extreme case (defined by an AOT of 1.15, mean plus 4 times standard deviation). According to the new Figure 1, we found 12 extreme dust storms with AOT>1.15 in these 15 years, most AOTs of these extreme events were below 2.0, only two (1 April**

**2013, AOT of 4.2, and 8 September 2015, AOT of 4.9) exceeded 4.0. This is now emphasized in the Introduction section, in the first paragraph, based on Figure 1.**

**PM10 time series are at all not useful, because Saharan dust storms are often lofted, whereas the Middle East storms often cover the entire lower troposphere from the surface up to 4 km height. During the Saharan dust storm on 1 April 2013 with AOT of around 4.0 (at noon according to AERONET), the maximum dust mass concentration was 1300 µg/m3, on 8 September 2015, it was 7600 µg/m3.**

…………………………………………………………………………………………………..

**Reviewer #1 (minor revisions):**

I found some parts difficult to follow. There were results mentioned in the methodology part and vice versa. It would be beneficial if you could work on the structure of the manuscript.

**Guided by your detailed hints (below), we tried to better separate methodological points (went into section 2) and results (went into section 3)**

page 2, line 35 Consider including Preißler et al. (2011).

**Done (Introduction), now cited. We carefully checked the literature months ago, but obviously overlooked this very relevant paper. Sorry!**

page 4, line 5 What output does MODIS provide for AOT>5? Are those data points set to 5? Please include this information here (it is mentioned in the results section).

**We explain it better now (section 2 as well as in the result section). Yes, there are many data points just set to 5.0 in the MODIS data base (indicating that the true AOT was larger than 5.0). However, we used all available data in the averaging, so also these 5.0 values….. Consequently, the true AOT is underestimated in this way. Even if we would exclude all 5.0 values, the AOT would be biased (strong underestimation of the true area mean AOT).**

page 4, line 5 move "On 8 September, this value was frequently exceeded." to results section

**Improved**

page 4, line 12-13 move "We found ... September)." to results section

**Improved**

page 5, line 1-3 move "During the strong dust ... can be neglected." to results section

**Improved, but at some places in the uncertainty discussion in section 2 we need to mention already some dust results (from section 3) to make clear that other effects (caused, e.g., by anthropogenic or marine aerosols) are negligible and thus do not introduce uncertainties in this specific case with the enormous dust load.**

page 5, line 25 Could you please explain why you chose a point 160 km East of Limassol over the sea as trajectory end-point? Is it because the trajectories ending over the island failed to capture the observed dust transport?

**According to MODIS, the thickest dust plumes occurred east of Cyprus. For this area we computed the trajectories. We state that now more clearly in section 3. But, yes, the HYSPLIT trajectories were really bad for Limassol. We leave out to mention that to avoid any confusion here… The trajectories are just shown to provide an idea about the main dust advection path.**

page 5, line 25 Do you mean between 8 and 9 UTC on 8 September? How do you know without lidar observations and radiosounding profiles at 6 and 12 UTC?

**Yes, we mean 8 September, and we make the statement more specific (see section 3.1): The peak dust front reached Limassol *at ground* between 8–9 UTC on 8 September (see photographs below taken briefly after the arrival of the dust front). So, we know it from our own observation (by eye).**

Technical corrections

**We checked all the points (below) and corrected them (100%)**

abstract Please provide full name for MODIS and EARLINET.
page 1, line 10 "close to 8000 _g/m3" or "near 8000 _g/m3"
page 1, line 21 remove ">"
page 3, line 8 change "belongs to" to "is part of"
page 3, line 16 remove last "and"
page 5, line 20 change "a coarse idea information" to "a rough idea" or "an impression"
page 5, line 22 _gure and caption give arrival height of 2.5 km, text gives 1.5 km for
lower dust layer
page 5, line 23 add space after "(RH)"
page 5, line 20-25 I _nd it hard to follow this part due to a number of parentheses (and at
least one too many if I'm not mistaken). Please rephrase, incorporating
the information in parentheses in the text.
figure 5, caption change "at the roof" to "from the roof"
figure 6, caption move everything from "The AOTs are..." to main text
figure 6, map in panel c increase font size, it is very hard to read
page 8, line 15 "Kandler et al., 2011" in parentheses
page 9, line 4 use either "wrong visibility estimations" or "a wrong visibility estima-
tion"
page 9, line 4 change "unusual very" to "very unusual" or just "unusual"
page 9, line 7-8 there is a verb missing after "... so that the..."
page 9, line 7 change "volumne" to "volume"
page 9, line 23 change "visbility" to "visibility"
page 9, line 31 change "oberved" to "observed"
page 10, line 17 change "A more ... pro_les" to "More ... profiles"
page 11, line 8 change "keeping" to "taking"
page 11, line 16 change "a variety optical" to "a variety of optical"
page 11, line 20 replace "strength" by "intensity"
page 11, line 27 change "improved" to "improve"
page 11, line 34 remove "profile"

**Improved**

References
Preißler, J., F. Wagner, S. N. Pereira, and J. L. Guerrero-Rascado (2011), Multi-instrumental observation of an exceptionally strong Saharan dust outbreak over Portugal, Journal of Geophysical Research: Atmospheres, 116, D24,204, doi: 10.1029/2011JD016527.

**Included (Introduction section)**

…………………………………………………………………………………………………………………………

**Reviewer #2 (minor revisons):**

Minor comments and notes

• 3/29: 'The uncertainties in all the optical properties…': Here, one of the papers by Gasteiger and colleagues should be cited who performed a lot of quite detailed investigations on this topic (desert dust, volcanic ash).

**We include both references in Section 2.1, … Gasteiger 2011a,b. The reviewer is right. There are not so many papers dealing with irregularly- shaped (desert or volcanic) dust. So, we should cite these two Gasteiger papers to stimulate even more modeling in this field.**

• 4/12: '± 50': Unit is missing.

**Improved**

• 5/11: Fig. 1: Please mark Limassol in at least one of the panels

**Done, in all panels Limassol is now indicated by a red point.**

• 5/29: Why are the lidar measurements restricted to day-time measurements: lack of personnel? I assume the system does not run unattended?

**We provide more information regarding the measurement time periods (section 3). But all the measurements were performed until about 21 UTC, … which is midnight in Cyprus! Nighttime observations were always performed to have at least 2-3 hours for Raman extinction profiling.**

• 6/8: 'and 600 µg/m3…'. When averaging over 3 hours this is certainly a lower limit. From Fig. 3a it can be seen that during the first hour the concentration was significantly larger. So it was indeed a strong event!

**We checked all this. We now present only mean values for 7 September 19-20 UTC (throughout the paper). And the mass concentrations for the 18-19 UTC (first hour) are not higher. After arrival of the dense dust layer below 1000m the signals above 1000m got strongly attenuated which gives the impression in the color plot of the range-corrected signals that there is less dust.**

• 7/17: 'Therefore the area mean values…' What does this mean? Is it – due to the fact that the maximum retrievable values might be exceeded – sort of an estimate of the lower limit?

**We explain the averaging procedure now in section 2 and section 3. We include all available data, even the ones set to 5.0 (which indicates..the true AOT was larger), in the averaging. That means, the computed area mean AOT value is clearly lower than the true area mean AOT. Even if we exclude the 5.0 values, the averaging would lead to underestimation, because we leave out to consider the high AOT values at all.**

• 8/16: Paragraph starting with 'To check…': I doubt that it is possible to determine a relationship between PM10 and TSP taking into account the uncertainties of the contributing variables/measurements (during this episode), and I don't understand the message of the related discussion. The authors found an 'excellent agreement' with Kandler's values. On the other hand they present arguments against the assumed values (stating that either cv,d or rvis is wrong). If however rvis is wrong this would feedback to the estimated extinction coefficient and the mass concentration. Some additional explanations would be helpful to avoid possible confusion.

**We simplified the text and reduced the information content in section 3. The only essential point is that Kandler et al. (2011) found that the total suspended particle (TSP) mass concentration is about a factor of 1.2-1.5 larger than PM10, which is quite reasonable because PM10 only considers particles with diameters < 10 μm. And when we combine our visibility observation of 500m (leading to TSP mass concentrations of 10000 micrograms per m3) and the in situ measured PM10 was 8000 micrograms per m3 then this is a good and reasonable match with the findings of Kandler et al. (2011). We state that. Afterwards we discuss the problems with the routine visibility observations at the airports and the large uncertainties is the estimated visibility values leading to large uncertainties…. In this way all the complex information is better organized, and will produce less confusions .**

• 10/7: 'termine': typo.

**Improved**

………………………………………………………………………………………………….

**Reviewer #3 (major revisons):**

In general the authors replied to the majority of my comments adequately. However, I find that the discussion about the vertical distribution and the AOD on the 8th of September (the day with the heaviest dust load) is not convincing enough (see major comments for details). Although, I believe that the authors describe indeed an extreme event as indicated by the very low visibility and PM10 concentration during the 8th September, but I am not persuaded based on the observations that the AOD is larger than 5, while it can be 3 (of course the event is still extreme).That said I suggest again publication to ACP once the authors have taken into account the following comments.

**Yes, we agree! Therefore, we simplified the discussion (following the suggestion of the reviewer) and discuss two scenarios, only. We assume a well-mixed dust layer up to 800 m or 1500 m. With an extinction coefficient of 6 km-1 at all heights we end up with AOTs of 4.8 to 9. These value corroborate the MODIS data, which suggest AOT>5.0. We feel that such a speculative discussion is tolerable in a case of such a unique dust storm. All this is consistent with the radiosonde observations (at least not in contradiction with the radiosonde profiles).**

**However, we do not use AOT estimations by assuming that the main dust clouds were within the lowest 300 m. This was the third scenario suggest by the reviewer. This gives unrealistically low AOTs (of the order of 2-2.5). On the other hand, we never observed such shallow dust layers with the EARLINET lidar over Limassol from 2010 to 2015…, or at other places with our lidars (Morocco, Cabo Verde, Barbados, Tajikistan, South Korea…). So, this appears too speculative.**

Major comment

5) Page 2, Lines 14-15: Although, there are not observations from the CUT- TEPAK AERONET sun-photometer during the event, there are observations (at least Level 1.5) from other sun-photometers in the region like Agia Marina_Xyliatou and SEDE_BOKER. It is important to compare MODIS AOD retrievals with AERONET values in order to establish how good there are in the case of large AODs (>2). This is important, especially, from the moment that you are using MODIS AOD over Cyprus in Section 3.1.

**We agree! As a consequence, we used the AERONET data Agia Marina (30 k m west of Nicosia) and compared them with the MODIS observations (area mean, 50 km radius) for the full 7-12 September period.  We found good agreement, when taking the suggest uncertainty of 0.15xAOT into account, and when we also took into account the orographic aspects (changing heights above sea level…. over  the orographically detailed terrain in the area around Nicosia).**

**Then we tried to use Sede Boker AERONET data, but this did not work, there was a too low amount of quality-assured MODIS data… So, we checked the Weizmann Institute AERONET data, and were successful. Here are the findings for the Weizmann AERONET station:**

**8 Sep: MODIS: 5.0, AERONET: no observations**

**9 Sep: MODIS, 3.1 and 3.95, AERONET: 2.4-2.6**

**10 Sep: MODIS, 3.8, AERONET 2.4-2.8, and later … MODIS 2.9, AERONET: 2.0-2-4**

**Finally we checked the Limassol AERONET observations for the second largest dust storm (1 April2013)**

**MODIS: 4.5, AERONET 3.2-3.5 during the time of the overpass plus minus 30 minutes.**

**We briefly summarize these results regarding the MODIS uncertainties  in section 2.2**

We try to compare the MODIS observations with our own lidar observations (section 3.2). We do not include other stations. The dust outbreak was so inhomogeneous, what does it help to include other stations? The uncertainty of MODIS values is clear, about 0.15 times AOT.

REPLY: Indeed, this is the expected uncertainty based on statistical comparisons with AERONET. However, here we have a specific event and as such MODIS AOD retrievals can be from very good to very bad. The comparison of MODIS with the lidar confirms the good performace of the retrieval when the AOD is about 1 or less. This is a useful addition to the paper. On the other hand the 8th September, there are not lidar observations to compare with. However, this appears to be the most important day of the event (at least close to the surface, where observations are presented) with high AOD based on the MODIS images (as the surface is not visible). As I said in my first review it is important to validate the MODIS retrievals for high AOD as the validation against AERONET (e.g. Levy et al., 2013) is mainly for AODs less than 1 and simply we do not know the performance of MODIS AOD for the extreme cases with AOD greater than 1 or 2.

**We agree, the answer is given above!**

Finally, the estimation of the AOD using the surface observations and extrapolated vertically based on the radiosonde is very uncertain regarding the thickness of the layer. Just looking at both the potential temperature and the relative humidity I would say that the thickness is 0.8 km and not 1.5 km used in text, while there is an inversion at ~0.3 km, which could also be the top of the dust layer. Note that the lowest value is in agreement with the lidar observations of the 9th September. On the other hand the value of 0.8 km multiplied with 6000 Mm-1 gives an AOD of 4.8, which is closer to the values reported by MODIS to Figure 6. Briefly the MODIS AOD image of the 8th September should be given and the retrievals should be validated against the AERONET observations for this day. Ideally, this should be done also for the other days of the event 7-11 September (but it is not necessary given the comparison with the lidar, although there is significant time difference between the lidar observations and MODIS AOD, while for AERONET the time difference can be less than 30 min).

**We agree and follow the reviewer in section 3.1. However, we discuss the AOT in cases of well mixed layer up to 800m and up to 1500m as mentioned above (radiosonde discussion). There is no reason, to assume that a well-mixed layer up to 1500m is unrealistic. The radiosonde profiles are consistent with both assumptions. However, we need to be careful with the sonde profiles, the sondes were launched at Nicosia, more than 50km northeast of Limassol, and there was much less dust at Nicosia on 8 Sep, …. indicating weather conditions (and T and RH profiles) different from the ones at Limassol. We now explicitly say that we speculate here…**

**We leave out to discuss the third scenario (mentioned above). 300 m thick or shallow dust layers with huge dust extinction coefficients of 6 km-1 are simply too unrealistic, to our opinion. And if that was really the case, we are sure, that pilots of air planes landing at Larnaca around noon would have noticed that and would have reported that, … that there was a sharp shallow dust layer. But they just reported very low visibility of about 500m on 8 Sep during approaching the Larnaca airport and landing phase. This is in agreement with our visibility estimation (by using the phtographs). We leave out to mention the reports of the pilots (not needed).**

Minor comments

12) Page 3, Line 30: "A two-layer structure ... on 8 September," this is pure speculation. Either present observations or delete it.

**We think, page 5 (not page 3) is meant! Now we provide a more relaxed discussion, following the way suggested by the reviewer in Sect.~3.1.**

As already given above, we now include radiosonde information of 8 September. These data perfectly corroborate our 'speculation'. So, there was definitely a 1.5 km thick and well-mixed layer (with base at surface on 8 Sep, noon). And by using the visibility-related extinction coefficient of 6000 Mm-1 we end up with an optical depth of 9. All in all, lidar and radiosonde data are in perfect harmony for the entire period from 7 to 11 September. This motivated us to use the radiosonde profiles alone to estimate the layering on 8 September. We find, this is fully justified. So, we do now use the 8 September temperature and humidity profiles to explain the dust layering and that the dust was well mixed from the ground up to 1500 m height (as inidcated by the humidity profiles on 8 Sep) and caused and optical depth of 6-9. All this is now discussed in Section 3.2

REPLY: As I explained in my previous reply to comment 5 I disagree that the lower layer is well mixed and up to 1500 m. This is a crucial point for the AOD estimation during this day.

**What can we answer here! We understand the reviewer! But, it may be helpful to state in addition: Albert Ansmann (A.A., second author) launched more than 500 Vaisala radiosondes during lidar field campaigns during the last 20 years (lidar/radiosonde field sites), and feels experienced enough to interpret radiosonde profiles. For A.A. there is no doubt that the Nicosia sonde profiles corroborate well-mixed conditions up to 1500m on 8 September. This is clearly corroborated by the monotonic increase of RH up to 1500 m height of the 12 UTC sonde on 8 September. However, A.A. is careful enough not to trust the sonde profile data too much. We have to consider that the radiosonde data in the data base only show significant changes in the temperature and RH profiles (we would like to have the original profiles with very high vertical resolution), and that the launches were conducted far away from the lidar site and not exactly to the time when the sharp dust front crossed Cyprus, and differently in northern (Nicosia) and southern Cyprus (Limassol). This does not mean that we better avoid to use these radiosonde profiles. No! It means, we should just be very careful, and if we see consistency between lidar and radiosonde profile structures we should take that as an indication that the air mass characteristics seen by the radiosonde was obviously similar to the one above the lidar site.**

**But we got the point of the reviewer, and changed the discussion towards a more careful argumentation…**

13) Page 4, Lines 5-7: This is just a speculation, as you do not have any information about the vertical distribution of the dust. The authors should underline more this fact using a stronger word than assumption. The fact that the Troodos Mountains were not covered by dust means that possibly the dust layer was below 2 km, but it could be just 500 m (or lower) like the 9th September (Figure 5). On the other hand, Figure 6 shows through the backscatter coefficient that the layer was not homogeneous with the higher values at surface during the 9th September, so this could be the case for the 8th. At this point MODIS maps can give a hint about the area with the high AOD (even saturated at 5), while also the AOD observations from MODIS/Terra could be useful to check the temporal variability. Certainly, this result although plausible can not be one of the main conclusions of the paper as it is based on assumptions which can not be verified. Instead the authors could use the MODIS AOD.

**We checked corresponding MODIS-based AOT maps and concluded not to follow the reviewer in this point. These maps will trigger new questions because it is simply not easy to provide a rather dense quality-assured AOT data sets over the Med Sea and over Cyprus (a very crucial island for satellite remote sensing, with mountains up to 2000m, with bright desert and dark forest surfaces…, etc). The suggested maps are however only useful if the data set is dense (not too many gaps). We estimated that we need one week to carefully check all available MODIS data to produce such an AOT map. And at the end, we will have many yellow areas (indicating AOT>5.0) in the southern parts of Cyprus and over the Med Sea (to the south). What does that help??? It makes everything too complicated. So, we leave out to produce such a map. Maybe MODIS 'enthusiasts' will be triggered by our article to do that….**

As mentioned, we now include radiosonde information of 8 September….. These data indicate a well-mixed 1.5 km thick lower dust layer. And with surface extinction coefficients of 4000-6000 Mm-1 we end up with AOT of 6-9. And this also in agreement with MODIS. MODIS stops when the AOT is largere than 5. And there are many MODIS pixels (areas) with higher AOT than 5, as discussed in

section 3.2. MODIS does not provide data higher than 5. We use MODIS TERRA and AQUA data and do not see strong temporal variability…

REPLY: Similarly to my reply to comments 5 and 12, I disagree about the thickness of the lower layer and the fact that it is well mixed. In any case I do not think that it is difficult for you to add one more figure to the paper with the MODIS AOD of the 8th September, especially from the moment that you have the data already. It is just 10 min work to create it.

**10 minutes!!!  We have the feeling we need much more time because we are not MODIS retrieval experts,  and this map will then trigger new questions. And we do not think that such a map is needed. So, we leave out to present such a map.**

New comments

1)Page 1, Lines 20-22 and Page 2, Line 1: These are your results, so I do not think that it appropriate to start the paper with them, so please delete them. If it's not the case provide the references.

**We changed the text, but we need to start with the MODIS observations. We now analyzed MODIS observations from 2001-2015. These data allow us to immediately state that this 8 Sep 2015 dust storm was a unique event. We provide a reference for the MODIS data base and a new Fig. 1.**

2)Page 2, Lines 6-7: "However … decade." Where this statement comes from?

**This statement is now based on our 2001-2015 MODIS data analysis. The 8 September peak AOT is the largest of the last 15 years (see Fig.1).**

3)Page 4, Line 5: "On 8 … exceeded." I think it is better to move this sentence to Section 3.

**We changed this text part (section 2.2 on MODIS) completely. We provide an extended error analysis, including for cases with AOT>2.5.**

**4**)Page 4, Line 12: "+/-50". I think the % is missing.

**Improved**

5)Page 4, Line 31 and Page 5, Lines 1-3: "During … neglected." These are results and are better suited for the Section 3.

**No! In section 2.4 (on visibility observation) we discuss all error sources, and to make clear that we can neglect anthropopenic and marine aerosol contributions to the reduced visibilities, we need to provide the optical properties found during the reported dust storm.**

6)Page 5, Line 20: Clearly, I do not see what the HYSPLIT model adds on the paper, especially from the moment that MODIS images show the origin and evolution of the dust plume and the starting point of the backwards trajectory is not Limassol.

**The starting point is in a region with the densest dust plumes according to MODIS. We state that now. The HYSPLIT trajectories are just given (or better needed) to show the main air mass transport way. MODIS maps in Figure 1 do not show that so clearly.**

7)Page 6, Line 1-2: Why you do not present radiosondes also for the other dates of the event 7-11 September? This will further corroborate the connection of the layering seen by the lidar and the radiosondes.

**As mentioned several times above: The sondes provide a rough idea about the temperature und relative humidity profiles over Limassol. And our experience with sonde/lidar observations show that they are always in very good agreement. So, we do not think that we need the sonde profiles. Just to show that the RH and temperature profiles are in good agreement with the lidar profiles is not enough motivation for us to show them.**

**We usually like to show radiosonde profiles but in this case, we feel , it does not contribute much. And the radiosonde site is close to Nicosia, 70 km northeast of Limassol. This area was excluded from the main dust transport regime. Furthermore, the sondes were launched at 12 UTC or early, and our lidar observations are mainly done in the late afternoon and evening.**

8)Page 6, Line 8: Replace "exceeded already" by "reach".

**Improved**

9)Page 6, Line 18: AOT of what? The lidar? You have also the time series of MODIS in Figure 6.

**Improved …… AOT from lidar and MODIS observations….**

10)Page 6, Line 23: How the values of 8-10 km are estimated?

**We took these data from the routine visibility observations at the three airports. But these are rough estimates. We changed the text … visibility > 10 km is a more save statement.**

11)Page 7, Lines 2-9: In the model deficiences you are referring to cloud convection. However, in MODIS images there are not clouds in the source region.

**The clouds can be found on MODIS maps of 7 September, east of the region shown in the old Figure 1 (now Fig.2, eight satellite maps) . This map (to the east) will be shown in the second paper on modeling (Solomos et al.).**

12)Page 7, Line 25: Why there is not comparison of the lidar and MODIS the 7th September?

**The dust changes are large on 7 September, and the MODIS results are given for morning and noon hours, and the lidar observations were done in the evening. A comparison is useless in this case.**

13)Page 7, Lines 29-30: Once again, why you do not show the radiosondes?

**We find that it is sufficient to provide the general radiosonde results here. We already stated several times why we do not show radiosonde data.**

14)Page 7, Line 32: How the values of AOD for anthropogenic and marine aerosols are estimated?

**From our own long-term lidar and AERONET studies at Limassol, and by other AERONET observations over remote Ocean sites. We provide references and more explanations now.**

15)Page 11, Lines 17-18: "The highlight … observations." Sorry to disappoint you, but the most important day of the event the 8th September, there are not lidar observations. So, this sentence is not justified, please delete it.

**We changed the text accordingly**

16)Page 11, Line 18: According to your reply to my comment 15, I do not think appropriate to call your lidar state-of-the-art. Please delete this characterization.

**We removed 'state-of-the-art'**

17)Page 11, Line 22: "constellations" you mean "conditions"?

**Improved**

[revised manuscript text omitted]